# Generative vs Discriminative: Rethinking The Meta-Continual Learning

**Mohammadamin Banayeeanzade**,*   **Rasoul Mirzaiezadeh**\*,
**Hosein Hasani**\*,   **Mahdieh Soleymani Baghshah**

Department of Computer Engineering
Sharif University of Technology

m.banayeean@gmail.com,   mirzaierasoul75@gmail.com
hasanih@ce.sharif.edu,   soleymani@sharif.edu

## Abstract

Deep neural networks have achieved human-level capabilities in various learning tasks. However, they generally lose performance in more realistic scenarios like learning in a continual manner. In contrast, humans can incorporate their prior knowledge to learn new concepts efficiently without forgetting older ones. In this work, we leverage meta-learning to encourage the model to learn how to learn continually. Inspired by human concept learning, we develop a generative classifier that efficiently uses data-driven experience to learn new concepts even from few samples while being immune to forgetting. Along with cognitive and theoretical insights, extensive experiments on standard benchmarks demonstrate the effectiveness of the proposed method. The ability to remember all previous concepts, with negligible computational and structural overheads, suggests that generative models provide a natural way for alleviating catastrophic forgetting, which is a major drawback of discriminative models. The code is publicly available at https://github.com/aminbana/GeMCL.

## 1   Introduction

Deep learning methods have gained great success in a wide variety of tasks in various fields [26, 28]. However, in many real-world scenarios, where the data samples are neither i.i.d. nor available all at once, most of them lose performance on the older data points; a phenomenon known as catastrophic forgetting [13]. Continual learning, also called lifelong learning, aims to tackle this problem [48].

Animals as intelligent learning agents can easily learn a sequence of different tasks with minimum interference and forgetting [55, 12]. When facing new problems, the brain does not rewire all of the wirings [37, 5] but utilizes lifelong accumulated inductive biases to efficiently learn new concepts [9, 4]. In the field of cognitive and neuroscience, there exist two special cases of concept learning named prototypical and exemplar-based learning [44, 2, 59, 7]. Bayesian inference is another aspect of learning in the brain which aims to consider uncertainty [24, 31, 16, 4, 23]. In this paper, we integrate the hypothesis of generative concept learning and the Bayesian approach by considering a distribution over each concept. Figure 1 compares different characteristics and expressive power of the mentioned concept learning approaches. Inspired by the biological mechanism of learning, we suggest that the generative approach in category learning is an important principle for continual learning problems. Although the Bayesian approach requires proper priors about each concept,

---

*Equal contribution

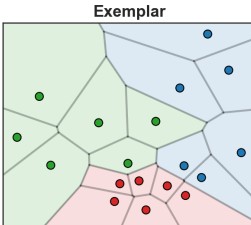 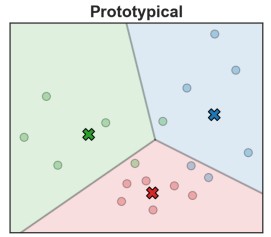 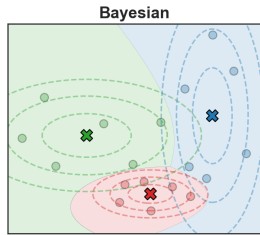

Figure 1: A simple schematic of exemplar-based learning (left), prototypical-based learning (center), and the proposed Bayesian approach (right). Exemplar-based learning suffers from high memory consumption and also the risk of overfitting, like 1-NN classifiers. Prototypical-based learning is memory efficient but also prone to underfitting. Our probabilistic approach enables using complex enough distributions over each category to capture class variations.

meta-learning can be used to provide data-driven prior knowledge or statistics from the previously encountered concepts similar to the biological intelligent systems [4].

Meta-learning is an endeavor to gather experience from solving related problems in order to solve new unseen ones more efficiently; which have been used to improve the learning process in different areas [19]. In this work, we employ meta-learning to derive proper inductive biases as well as a powerful feature extractor for a continual learning problem from a collection of similar continual learning problems. By taking advantage of these inductive biases, we formulate the category learning problem from a probabilistic generative approach. Our main contributions are as follows:

- We take motivation from cognitive and neuroscience about meta-continual learning in intelligent animals and especially make use of bio-inspired concept learning to eradicate catastrophic forgetting.
- We investigate the problem of discriminative approaches in continual learning and propose generative classifiers as a general way of mitigating catastrophic forgetting.
- Encouraged by generative classifiers, we propose a new discriminative classifier called PGLR, to reduce the inherent catastrophic forgetting problem of discriminative approaches.
- Existing prototypical learning is improved by adding extra statistics through the Bayesian approach.
- Different components of the proposed method can be implemented in a fully incremental setting which is useful in the shortage of data or computational resources.
- We achieve state-of-the-art accuracy with a notable gap compared to the existing methods.

## 2 Related Works

**Continual Learning:** Three main approaches have been proposed to address the forgetting issue [33]. (i) Replay-based methods rely on reusing samples from previous training tasks either by keeping them in an explicit episodic memory [30, 36, 6, 1] or generating pseudo-samples via a deep generative model [43, 53, 50]. (ii) Methods based on parameter isolation dynamically expand the model as the data for new tasks arrive by introducing task-specific weights[40, 57, 42] or masks[52, 58]. (iii) Regularization-based methods consider a probability distribution, typically the Gaussian family, over parameters and penalize the substantial deviations from the prior distribution learned through the earlier tasks. Taking a Bayesian approach will lead to an incremental posterior update which is naturally consistent with continual learning [32], while considering a point estimation for the parameters will result in a regularization term in the objective function [45, 60, 22]. Besides theoretical foundations, simplifying assumptions and approximations often make the regularization-based methods fail when the number of tasks is large and they are more prone to catastrophic forgetting compared to the first two approaches [49].

Compared to the first approach, merely storing low dimensional sufficient statistics of categories, makes our model extremely memory efficient and applicable even when there are thousands of tasks or categories. However, our method is more related to the second approach in the sense that we isolate class-specific parameters. Like the Bayesian methods [32], we also consider a distribution over

parameters. Nevertheless in our method, using generative classifiers and restricting the distribution to the class-specific parameters enables exact closed-form solutions with zero forgetting.

Most of the existing continual learning methods use a discriminative approach but there exist some prototypical or exemplar-based models which are related to our method in the sense that they utilize a generative approach for final classification [36]. However, they are not completely immune to forgetting and are also less memory efficient since they store a considerable number of high-dimensional input examples.

**Meta-Learning:**   Existing methods in the literature of meta-learning can be categorized into optimization-based [10, 35], black-box [41, 14], and metric-based [46, 47, 56] approaches. In this paper we use metric-based methods that meta-learn an embedding space in which a metric-based classifier performs well. Some methods called Bayesian meta-learning, approach the problem from a probabilistic point of view, obtaining a distribution over the problem-specific parameters, therefore capable of providing information about the uncertainty of a learned model [15, 11, 54]. We also provide a probabilistic framework capable of using various statistics of data and learnable prior distributions.

**Meta-Continual Learning:**   Both continual and meta-learning have mutual impacts on each other, introducing two new research branches [8]: Continual meta-learning [18, 21] and meta-Continual learning [17, 38, 20, 3]. In the second branch, various techniques from meta-learning are used to tackle catastrophic forgetting. Here, we especially focus on the settings in which explicit episodic meta-training on continual problems is used [20, 3]. OML [20] meta-learns a representation network to avoid interference between the continual tasks and reduce the catastrophic forgetting. ANML [3] also takes the same approach but instead of merely having a representation network, it generates a data-dependent mask that modulates the activity of neurons in the prediction network.

Motivated by them, we utilize episodic meta-learning to tackle the continual learning problem, but instead of having a discriminative approach, we propose a probabilistic generative method that is completely immune to forgetting. Our approach falls into metric-based meta-learning, as opposed to the prior methods that are optimization-based [20, 3].

## 3   Preliminaries

**Meta-Learning:**   In the meta-learning setting, we have access to a set of meta-training problems $\mathcal{D}_{\text{meta-train}}$ which help to learn a prior knowledge and thus generalize to unseen problems with data from $\mathcal{D}_{\text{meta-test}}$. Each problem is a pair of support and query sets, i.e. $(\mathcal{S}, \mathcal{Q})$, and labels from $\mathcal{D}_{\text{meta-train}}$ do not intersect with the labels in $\mathcal{D}_{\text{meta-test}}$. The mentioned prior knowledge can be encoded in the meta-parameters $\theta$ which is utilized by a problem-specific model parameterized by $\phi$ to solve a new problem instance. The overall learning procedure of the meta-parameters can be formulated as [11]:

$$\theta^{\star} = \underset{\theta}{\text{argmax}} \log P(\theta \mid \mathcal{D}_{\text{meta-train}}). \tag{1}$$

And the distribution of the problem-specific parameters, given a support set $\mathcal{S}$, can be formulated as:

$$P(\phi \mid \mathcal{S}, \mathcal{D}_{\text{meta-train}}) = \int_{\theta} P(\phi \mid \mathcal{S}, \theta) P(\theta \mid \mathcal{D}_{\text{meta-train}}) d\theta$$
$$\approx P(\phi \mid \mathcal{S}, \theta^{\star}) P(\theta^{\star} \mid \mathcal{D}_{\text{meta-train}}) \propto P(\phi \mid \mathcal{S}, \theta^{\star}). \tag{2}$$

While some works model this distribution, others use a point estimate as:

$$\phi^{\star} = \underset{\phi}{\text{argmax}} \log P(\phi \mid \mathcal{S}, \theta^{\star}). \tag{3}$$

**Meta-Continual Learning:**   A meta-learning scenario where each individual problem $(\mathcal{S}, \mathcal{Q})$ is continual, i.e. the support-set is observed continually, is called meta-continual learning. We consider support stream as $\mathcal{S} = \{\mathcal{S}_1, \mathcal{S}_2, ..., \mathcal{S}_t, ...\}$, in which $\mathcal{S}_t = \{(x_i^t, y_i^t)\}_{i=1}^{|\mathcal{S}_t|}$ and the $x_i^t$ and $y_i^t$ are the $i$th data point and the corresponding label in $\mathcal{S}_t$ respectively. Each $\mathcal{S}_t$ is accessible only at time-step $t$ due to continual learning restriction. The query-set $\mathcal{Q}$ is a set of samples sharing the same set of labels as those in $\mathcal{S}$. The objective here is to perform well on $\mathcal{Q}$ after sequentially observing all of

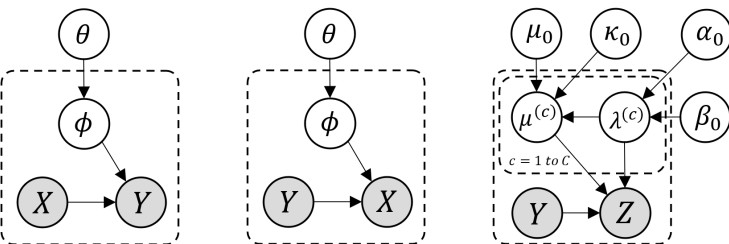

Figure 2: Graphical models for the discriminative (left), generative (center), and our (right) approach.

the samples available in $\mathcal{S}$. We use the notation $\mathcal{S}_X$ and $\mathcal{S}_Y$ for the data points and their labels in $\mathcal{S}$ respectively, and the same holds for $\mathcal{Q}_X$ and $\mathcal{Q}_Y$. Furthermore $\mathcal{S}_X^c$ shows the data points with the label $y = c$ available in $\mathcal{S}$, i.e. $\mathcal{S}_X^c = \{x | (x, c) \in \mathcal{S}\}$.

In this paper, we consider a model immune to catastrophic forgetting if the accuracy and decision boundaries of the model in continual learning and i.i.d. scenarios remain identical.

In the lifelong-learning literature, each $S_t$ is called a learning task which may contain the data points and corresponding labels for multiple classes. With this in mind, three standard benchmarks are introduced for evaluation: task-incremental, domain-incremental, and class-incremental. The latest benchmark is known as the most difficult one in which each class is only observed in a single task while at the test-time the model is expected to classify across all observed classes from all tasks [49]. However, our proposed method is completely agnostic to the order in which data samples arrive and produces the same model, even in an extreme setting that we refer to as fully incremental. In this setting, only one sample is available at each time step and each data point is visited exactly once. In fact, the only assumption in this work is that the samples with the same label in $\mathcal{S}$ are i.i.d., namely $P(\mathcal{S}^c) = \Pi_{x \in \mathcal{S}^c} P(x_i)$.

## 4 Method

Although recently meta-learning representations for continual learning have been introduced to alleviate the catastrophic forgetting [20, 3], these methods still greatly suffer from catastrophic forgetting. One probable reason is that they have constructed their methods based on discriminative approaches. Meanwhile, it is reported that the human brain benefits from a generative approach in category learning [44, 2, 59, 7] and this raises the question of whether humans capabilities in continual learning stems from their generative approach to the problem. While further studies by the neuroscience community are needed to investigate the exact relationship between biological category learning and the forgetting concept, here we provide some theoretical insights which shed light on the topic. In the following subsections, we elaborate on this by first discussing the drawbacks of the existing meta-continual learning methods which originates from their discriminative nature. Then, we show how the generative approach can resolve the catastrophic forgetting problem of meta-continual learning and finally, the proposed generative model for continual learning is presented.

### 4.1 Generative vs. Discriminative Approach

Probabilistic classifiers generally can be categorized into (1) Discriminative and (2) Generative models where they respectively model $P(Y \mid X)$, and $P(X \mid Y)$, demonstrated by Figure 2.

Given a classification problem $\mathcal{S} = \{\mathcal{S}_X, \mathcal{S}_Y\}$, the discriminative form of Equation 2 is:

$$P(\phi \mid \mathcal{S}_X, \mathcal{S}_Y, \theta^\star) \propto P(\mathcal{S}_Y \mid \phi, \mathcal{S}_X) P(\phi \mid \theta^\star). \tag{4}$$

In continual-learning settings, there is partial access to $\mathcal{S}$ at each time step, therefore the learning agent is prone to catastrophic forgetting. To clarify this, assume a class-incremental setting, where at each time step, the data of only one class is accessible but all the parameters are updated. Here, a trivial solution for each time step $t$ (assuming uninformative prior $P(\phi \mid \theta^\star)$) is a model that assigns high probability to the currently available label even for samples from other categories. Therefore, the solution for each time step works poorly on other time steps.

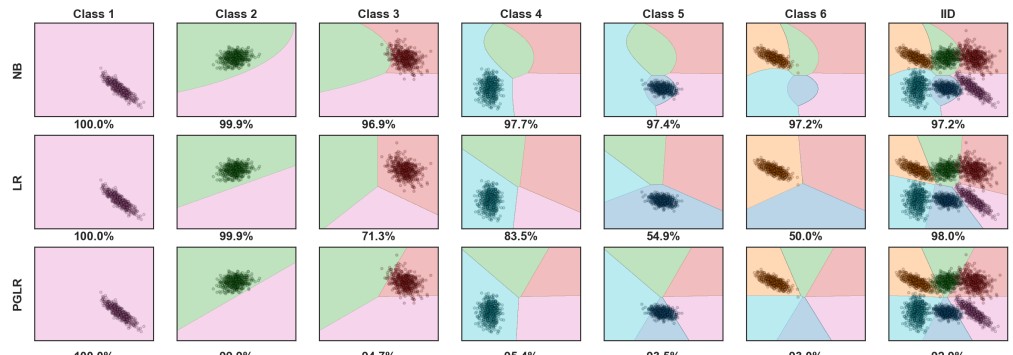

Figure 3: Comparison of discriminative and generative approaches on a toy dataset in continual and i.i.d. scenarios. The first and second rows show the results of a Gaussian naive Bayes classifier and logistic regression as generative and discriminative classifiers respectively. The last row shows the performance of the described PGLR model. The average accuracy on all observed classes, current class samples, and the decision boundaries of each model are shown from the first to the sixth column. The last column shows the performance of each model in the i.i.d. batch learning setting.

The other choice for classifier formulation is the generative model which can be formulated as follows:

$$P(\phi \mid \mathcal{S}_X, \mathcal{S}_Y, \theta^\star) \propto P(\mathcal{S}_X \mid \phi, \mathcal{S}_Y)P(\phi \mid \theta^\star). \tag{5}$$

In this approach, there is no interference between learning of different classes and hence it does not suffer from forgetting. In fact, at each time step, only the parameters of those classes visited at that time step are changed and parameters of other classes are neither utilized nor updated. With this in mind, the following theoretical statements are presented:

**Statement 1.** In the class-incremental scenarios, the generative classifiers are immune to forgetting, given any class conditional parametric distributions.

**Statement 2.** In the fully incremental scenarios, the generative classifiers are immune to forgetting, if the class conditional distributions have incrementally-updatable sufficient statistics. In the case of Bayesian inference, this condition is satisfied if class conditional distributions have conjugate priors.

To provide more insights, here we compare Gaussian naive Bayes classifier (NB) and logistic regression (LR) as common generative and discriminative classifiers on a continual learning problem. We also implement a pseudo-generative version of the LR classifier (PGLR) to reduce the forgetting problem which is an inherent characteristic of the LR classifier. In the PGLR model, only the class-specific parameters are updated during optimization and the others remain frozen (see supplementary materials). Class-specific updates make the PGLR classifier one step closer to generative models and prevent interference of tasks during learning. As shown in Figure 3, unlike the discriminative model, the generative models are immune to forgetting and there is no difference between the accuracy and decision boundaries of the continual (sixth column) and i.i.d. (seventh column) scenarios.

## 4.2 Proposed Model

As mentioned, generative classifiers are suitable for continual learning problems. However, the main challenge here is that training a reasonably expressive feature extractor is not straightforward in a fully generative manner. Hence, we employ meta-learning to tackle this challenge and propose a generative meta-continual learning (GeMCL) method. Like OML [20], we use a feature extractor network $f_\psi(X) : \mathcal{X} \to \mathbb{R}^d$, parametrized by $\psi$, that embeds the input into a proper $d$-dimensional representation space. However, inspired by the biological mechanism of concept learning, we incorporate a generative Bayesian classifier instead of using a discriminative one. Indeed, to have a tractable generative model, a proper representation space is first prepared by the embedding function and the optimal weights for this function are obtained via a meta-training procedure. Then, the hyper-parameters of the generative model in this space are computed based on $\mathcal{D}_{\text{meta-train}}$.

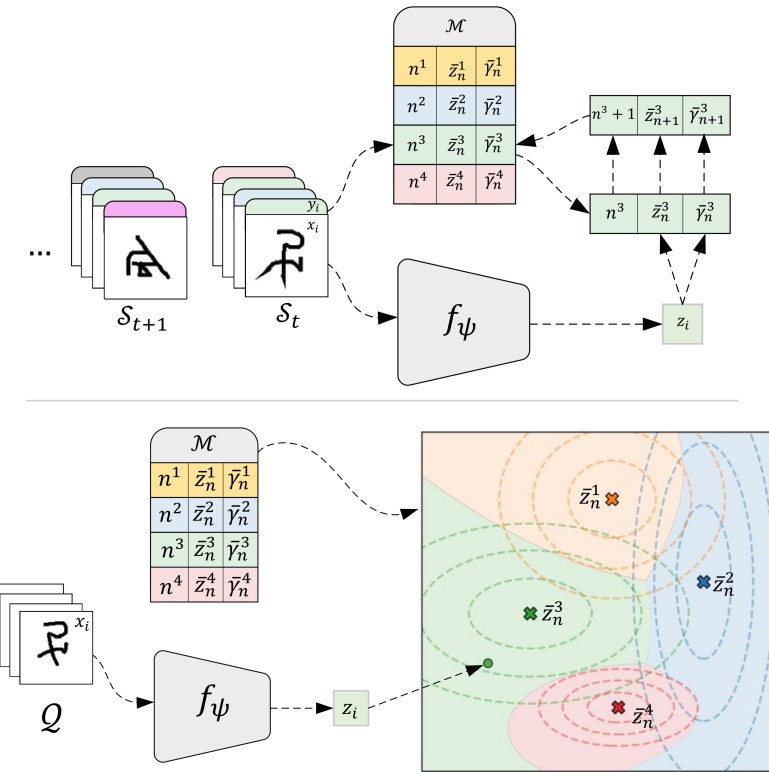

Figure 4: The overall procedure of a continual episode in GeMCL, demonstrating learning of the classifier parameters as the support data arrives (top), and classification of query samples given the learned class distributions (bottom).

In the meta-test phase, the generative model on the test classes is formed using incrementally arriving samples in the embedding space and the meta-learned hyper-parameters. In this phase, same as the previous works [20, 3], the embedding parameters $\psi$ are fixed. Moreover, in the prior continual learning works, it was investigated that the early representation layers are less responsible for forgetting, and remain nearly unchanged even after training for new tasks [34]. In fact, even a pretrained feature extractor along with a generative classifier outperforms the previous meta-continual models that are discriminative. We will investigate this more in Section 5.

### 4.2.1 Bayesian Generative Model

To derive a generative model, we need to choose a parametric family of distributions in the $d$-dimensional embedding space. We choose a Gaussian distribution to model each $P(z \mid Y = c)$ where $Z = f_\psi(X) \in \mathbb{R}^d$. In this generative model, we introduce $\mu^c \in \mathbb{R}^d$, $\lambda^c \in \mathbb{R}^d_+$ as the mean and precision of the class conditional Gaussian distributions respectively:

$$\forall_{1 \leq c \leq C} P\left(z \mid Y = c, \mu^{1:C}, \lambda^{1:C}\right) = \mathcal{N}\left(z \mid \mu^c, diag(\lambda^c)^{-1}\right), \tag{6}$$

where $C$ denotes the number of observed classes, $\mu^{1:C} = \{\mu^1, \cdots, \mu^C\}$ and $\lambda^{1:C} = \{\lambda^1, \cdots, \lambda^C\}$ are the set of parameters for Gaussian distributions. To obtain a Bayesian perspective, we consider a Normal-Gamma prior for the mean and variance of each class:

$$P\left(\mu^c, \lambda^c \mid \alpha_0, \beta_0, \kappa_0, \mu_0\right) \propto \prod_{j=1}^{d} \mathcal{N}\left(\mu_j^c \mid \mu_0, (\kappa_0 \lambda_j^c)^{-1}\right) Ga\left(\lambda_j^c \mid \alpha_0, \beta_0\right), \tag{7}$$

where $\alpha_0, \beta_0, \kappa_0, \mu_0$ are the hyper-parameters of the generative model. Now, given the support set of each class in the embedding space, i.e. $f_\psi(\mathcal{S}_X^c) = \{f_\psi(x) \mid \forall x \in \mathcal{S}_X^c\}$, the posterior and the predictive distributions are in the form of Normal-Gamma and student's t-distribution respectively:

$$P\left(\mu^c, \lambda^c \mid f_\psi(\mathcal{S}_X^c), \alpha_0, \beta_0, \kappa_0, \mu_0\right) \propto \prod_{j=1}^{d} \mathcal{N}\left(\mu_j^c \mid \mu_{n,j}^c, (\kappa_n^c \lambda_j^c)^{-1}\right) Ga\left(\lambda_j^c \mid \alpha_n^c, \beta_{n,j}^c\right), \quad (8)$$

$$P\left(z \mid f_\psi(\mathcal{S}_X^c), \alpha_0, \beta_0, \kappa_0, \mu_0\right) = \prod_{j=1}^{d} t_{2\alpha_n^c}\left(z_j \mid \mu_{n,j}^c, \frac{\beta_{n,j}^c(\kappa_n^c + 1)}{\alpha_n^c \kappa_n^c}\right), \quad (9)$$

where $\alpha_n^c, \beta_{n,j}^c, \kappa_n^c, \mu_{n,j}^c$ are the posterior parameters after observing $n^c$ samples of the class $c$. More specifically, by considering the first two moments of the Gaussian distribution as the incremental sufficient statistics, we can formulate the mentioned parameters as:

$$n^c = |S^c|, \quad \bar{z}_{n,j}^c = \frac{1}{n^c}\Sigma_{z \in f_\psi(\mathcal{S}_X^c)} z_j = \frac{(n^c-1)\bar{z}_{n-1,j}^c + z_{n,j}^c}{n^c},$$

$$\bar{\gamma}_{n,j}^c = \frac{1}{n^c}\Sigma_{z \in f_\psi(\mathcal{S}_X^c)} z_j^2 = \frac{(n^c-1)\bar{\gamma}_{n-1,j}^c + (z_{n,j}^c)^2}{n^c},$$

$$\kappa_n^c = \kappa_0 + n^c, \quad \mu_{n,j}^c = \frac{\kappa_0 \mu_0 + n^c \bar{z}_{n,j}^c}{\kappa_n^c}, \quad (10)$$

$$\alpha_n^c = \alpha_0 + \frac{n^c}{2}, \quad \beta_{n,j}^c = \beta_0 + \frac{n^c}{2}\left(\bar{\gamma}_{n,j}^c - (\bar{z}_{n,j}^c)^2\right) + \frac{\kappa_0 n^c (\bar{z}_{n,j}^c - \mu_0)^2}{2(\kappa_0 + n^c)}.$$

Therefore, at the test time, it suffices to merely store the first and the second-order moments and compute the above posterior parameters for each class separately (refer to Figure 4). These moments can be updated incrementally as each sample arrives, and consequently, this model has zero forgetting based on Statement 2.

It is noteworthy that all continual learning algorithms have to somehow support the new incoming classes, hence they have the obligation to expand the classifier parameters. While some of them mention this issue directly [36], others may consider an upper bound on the number of classes [20]. Compared to their methods, our approach has no assumption on the maximum number of incoming classes, making it flexible for unknown length sequence of classes.

Using Equation 3, we also propose a method called GeMCL-MAP, which uses MAP estimation for the mean and precision of the Gaussian distributions instead of the predictive distribution:

$$\mu_{\text{MAP}}^c, \lambda_{\text{MAP}}^c = \underset{\mu^c, \lambda^c}{\operatorname{argmax}} P\left(\mu^c, \lambda^c \mid f_\psi(\mathcal{S}_X^c), \alpha_0, \beta_0, \kappa_0, \mu_0\right) = \mu_n^c, \frac{\alpha_n^c - \frac{1}{2}}{\beta_n^c}$$

$$P\left(z \mid f_\psi(\mathcal{S}_X^c), \alpha_0, \beta_0, \kappa_0, \mu_0\right) \approx \mathcal{N}\left(z \mid \mu_{\text{MAP}}^c, diag(\lambda_{\text{MAP}}^c)^{-1}\right). \quad (11)$$

In order to adjust the hyperparameters for the prior distribution, we chose $\mu_0 = 0$ and $\kappa_0 = 0$ to provide a flat prior distribution for $\mu$. $\alpha_0, \beta_0$ are meta-learned as described in the next section.

### 4.2.2 Meta-Training And Meta-Testing

Following the previous meta-learning works, we first use episodic meta-learning to train the parameters of the embedding network. Although we consider a generative model in the embedding space, in the meta-training phase the embedding function itself is trained using a discriminative objective to avoid trivial solutions like $\forall x f_\psi(x) = c$. So using the generative forms presented in the previous section, we write the cross-entropy loss for a single episode as:

$$\mathcal{L}(\psi) = -\Sigma_{(x,y) \in \mathcal{Q}} \log P\left(y \mid f_\psi(x), f_\psi(\mathcal{S}_X), \mathcal{S}_Y, \alpha_0, \beta_0\right)$$

$$= -\Sigma_{(x,y) \in \mathcal{Q}} \log \frac{P(f_\psi(x)|y, f_\psi(\mathcal{S}_X), \mathcal{S}_Y, \alpha_0, \beta_0)}{\Sigma_{\hat{y}=1}^{C} P(f_\psi(x)|\hat{y}, f_\psi(\mathcal{S}_X), \mathcal{S}_Y, \alpha_0, \beta_0)} = -\Sigma_{(x,y) \in \mathcal{Q}} \log \frac{P(f_\psi(x)|f_\psi(\mathcal{S}_X^y), \alpha_0, \beta_0)}{\Sigma_{\hat{y}=1}^{C} P(f_\psi(x)|f_\psi(\mathcal{S}_X^{\hat{y}}), \alpha_0, \beta_0)}$$

$$(12)$$

Here, $\alpha_0$ and $\beta_0$ are two hyperparameters that define the shape of prior Gamma distribution for the precision parameter. During the meta-training, we set them to large values to better regularize the contractile behavior of embedding samples by imposing high variance on the embedding space of each category. Before the meta-testing phase, we calculate within-category variances of meta-train data embeddings and then use the obtained distribution to infer the amount of $\alpha_0$ and $\beta_0$ by maximum likelihood estimation (see supplementary materials for more details).

Finally, in the meta-testing phase, we use estimated $\alpha_0$ and $\beta_0$ for the prior distributions and update the class-specific posterior parameters as the support stream arrives. Then, to classify a query data point, we use $\hat{y} = \underset{y}{\mathrm{argmax}} P\left(f_\psi(x) \mid \alpha_n^y, \beta_n^y, \mu_n^y, \kappa_n^y\right)$ as the predicted label.

# 5    Experiments

We have performed our experiments on Omniglot [27], Mini-ImageNet [51], and CIFAR-100 [25] datasets. For all these datasets, we used the backbone proposed in [46] with an input size of $84 \times 84$, which uses less than 115K parameters compared to more than 2.9M ones in the prior works [20, 3]. This backbone is meta-trained to a maximum of 20K for Omniglot and 30K for Mini-ImageNet, and CIFAR-100 datasets and the best performing model is selected using early stopping. The training is done with a learning rate of 0.001, decaying to half every 0.1 of the training length.

The **Ominglot** dataset is commonly used in the meta-learning literature, containing 1623 classes of handwritten characters each with 20 samples. We use 763 and 200 classes for meta-train and meta-validation respectively, and others for meta-test. The reported experiments are done using 20-way 5-shot train episodes and 200-way 15-shot validation episodes with 15 and 5 query samples respectively.

The **Mini-ImageNet** dataset is a subset of the ImageNet [39], originally proposed for evaluating few-shot learning algorithms [51]. The dataset consists of 100 classes with 600 samples for each class and is divided into 64, 16, 20 classes for train, validation, and test meta-phases respectively. For this dataset, we used 20-way 10-shot and 20-way 30-shot train and validation episodes respectively with 30 query samples for both. We apply horizontal flip and random crop augmentations during meta-training.

The **CIFAR-100** dataset, contains 100 classes with 500 train and 100 test samples for each class. Following a recent work [29] we use 70 and 30 classes for meta-train and meta-test phases respectively. We use 10-way 10-shot episodes with 30 query samples for meta-training on this dataset.

For further insights, we introduce the following baselines in addition to the mentioned meta-continual learning methods (OML and ANML).

**Meta-Trained Logistic Regression (MTLR):** We implement a fully discriminative baseline similar to OML in methodology but incorporating the simple backbone used in the other baselines. Moreover, some implementation details of OML are omitted in this baseline, to purely compare different aspects of our proposed methods (refer to the supplementary materials for details).

**Pseudo-Generative Logistic Regression (PGLR):** This method was introduced in Section 4.1 to limit the amount of forgetting in the discriminative settings. This baseline can be obtained by applying two modifications to MTLR: (i) To compute the logits, the class-specific weights are normalized before dot-production. (ii) During the optimization, instead of updating all weights of the classifier head, only the weights corresponding to the available sample are updated (see supplementary materials).

**Prototypical:** This method was originally introduced for few-shot classification [46]. Here, we employ this approach for the meta-continual learning scenarios. Indeed, this model is a special case of our proposed method by choosing a unit-variance Gaussian model, and completely omitting the prior distributions in Section 4.2.1 and merely using the maximum likelihood estimation instead.

**Pretrained:** To clarify the impact of meta-training on different models, we pretrain the same backbone and replace it with the meta-trained feature extractors in some experiments.

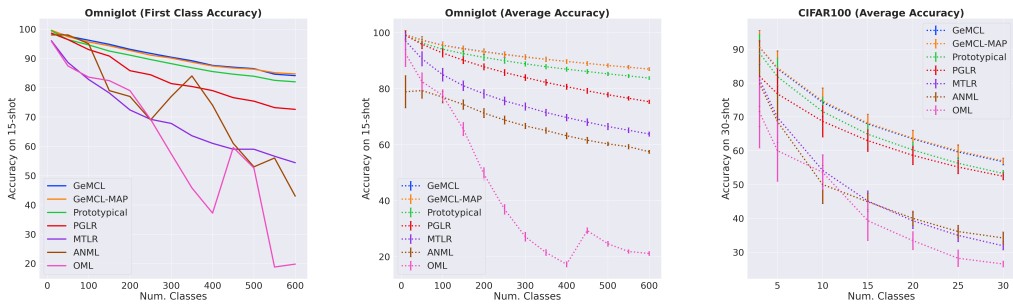

Figure 5: Effect of increasing the number of classes in Omniglot dataset measured based on the first seen class accuracy (left) and the overall accuracy (middle). Similar behavior for the average accuracy can be observed in CIFAR-100 dataset when the number of classes increases (right).

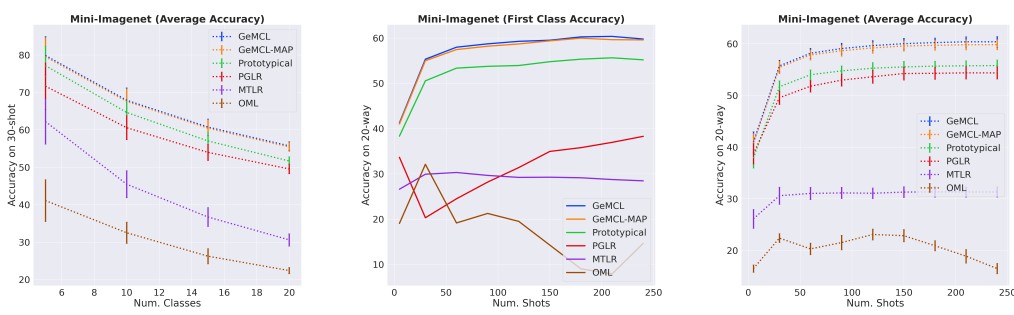

Figure 6: Effect of increasing the number of classes (left) and the number of samples per class (middle and right) in Mini-Imagenet dataset. The latter effect is measured on both the overall performance (right) and the performance on the first seen class (middle).

## 5.1 Results

In Figure 5 and 6, we evaluate our baselines, proposed methods, and two previous meta-continual methods from different perspectives. As shown, GeMCL outperforms OML and ANML by a large margin on both datasets. This margin grows as the length of continual problems increases. In fact, when observing more classes, the catastrophic forgetting causes a more notable performance drop in the discriminative models. Comparing the first class accuracy during test episodes, gives even better insights on the forgetting issue, as the earlier seen samples are more likely to be forgotten. Specifically, the comparably small performance loss of the generative models (GeMCL and Prototypical) is due to generalization issues rather than forgetting. Another way of enlarging the problem size is by observing more samples for each class. While more support data is expected to enhance the performance, the forgetting issue causes lower accuracy in OML. However, as the generative models do not suffer from forgetting, they perform even better in higher shots. Note that ANML and OML are very sensitive to the value of the inner-learning rate and tune it for each different episode setup (see supplementary materials). This results in selecting significantly small learning rates for longer episodes which trades better learning of each individual class for lower forgetting.

The prototypical baseline performs pretty well compared to the discriminative models, but its lower performance compared to GeMCL shows the effect of using additional statistics and trainable prior distributions, especially with more shots or classes the performance gap increases. It is worth mentioning that the PGLR model also considerably resists forgetting while being less sensitive to learning rate value. This reflects the potential benefits of the proposed pseudo-generative approach compared to discriminative ones.

Table 1: The performance of the MTLR (discriminative), PGLR (pseudo-generative), and GeMCL (generative) methods in their original meta-learning approach compared with their performance when using pretrained feature extractor. All of the models are evaluated on Mini-Imagenet with 20-way 100-shot settings and Omniglot dataset with 600-way 15-shot settings.

| Dataset | Mode | Head for Pretrained Backbone | | | Meta-train | | |
|---|---|---|---|---|---|---|---|
| | | MTLR | PGLR | GeMCL | MTLR | PGLR | GeMCL |
| Mini-Imagenet | Continual | $32.8_{\pm 3.1}$ | $54.2_{\pm 1.0}$ | $55.9_{\pm 1.1}$ | $31.1_{\pm 1.1}$ | $53.2_{\pm 1.1}$ | $\mathbf{59.2}_{\pm 0.9}$ |
| | IID | $\mathbf{60.9}_{\pm 1.0}$ | $55.8_{\pm 1.0}$ | $55.9_{\pm 1.1}$ | $48.9_{\pm 1.1}$ | $57.0_{\pm 0.9}$ | $59.2_{\pm 0.9}$ |
| Omniglot | Continual | $23.6_{\pm 0.7}$ | $61.0_{\pm 0.8}$ | $75.9_{\pm 0.8}$ | $61.9_{\pm 0.7}$ | $72.8_{\pm 0.6}$ | $\mathbf{86.9}_{\pm 0.6}$ |
| | IID | $77.8_{\pm 0.7}$ | $74.2_{\pm 0.6}$ | $75.9_{\pm 0.8}$ | $82.9_{\pm 0.5}$ | $86.8_{\pm 0.5}$ | $\mathbf{86.9}_{\pm 0.6}$ |

The i.i.d. and continual settings are compared in Table 1, showing that generative approaches are more robust to accuracy fall in continual learning. The performance of GeMCL is also superior to most of the baselines even in the i.i.d. settings. Furthermore, The results of using a pretrained feature extractor compared to meta-training show that GeMCL properly utilizes meta-learning to achieve better performance, which is not the case for MTLR and PGLR in some settings.

# 6 Discussion

In the present study, inspired by neuroscience, we propose a probabilistic generative classifier that is completely immune to catastrophic forgetting. Meta-learning makes training a deep generative classifier feasible and provides a rich feature extractor as well as prior knowledge about hyperparameters. Using this feature extractor, the proposed lightweight generative classifier achieves high accuracy for new unseen continual learning problems. It is noteworthy that our proposed method meta-learns hyperparameters on the meta-train dataset, and uses the same values for all meta-test episodes, while the optimization-based methods require inner learning rate tuning for various settings at test time. Moreover, we extend our approach to discriminative classifiers by introducing a pseudo-generative model which is not only less sensitive to inner-learning rate but also notably more robust to forgetting.

Our proposed method is efficient in different aspects: First, the employed backbone is light and overall computations are negligible compared to the current deep learning methods or other meta-continual learning approaches. Opposed to the previous meta-continual methods our proposed method does not use second-order optimization and is more computationally efficient by design. Second, our method is scalable and memory-efficient in the sense that it does not store data samples, in contrast to the memory-based continual learning methods [30, 36].

The proposed method has three main components which are basically inspired by neuroscience. First, the generative approach makes our method immune to forgetting. Second, meta-learning facilitates generative continual learning by providing a two-stage learning framework: slow learning in which backbone parameters and prior distributions are meta-trained, and fast learning in which class-related parameters are learned. Third, the Bayesian approach improves the performance and interpretability of the model by considering uncertainty.

In this paper, we have focused on solving some principal problems with existing meta-continual methods. However, there still exists some shortcomings which need to be addressed. These methods require a meta-train dataset to train a good embedding network which is not always available in some applications. Additionally, during the meta-testing, the embedding function does not adapt to new samples. This property offers protection from catastrophic forgetting, but it reduces the forward and backward transfer capabilities which are additional desired properties for continual learning methods. Furthermore, freezing the backbone may reduce the generalization capability that affects the overall performance, especially in scenarios that the number of continual tasks is limited and forgetting is not a major issue. Enabling further training of backbone during the meta-test phase could be a valuable direction for future studies. Especially, combining the other continual learning methods with the proposed method can offer forward and backward transfer in the meta-test phase and provide a desired trade-off between catastrophic forgetting, memory usage, and knowledge transfer.

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
