# Supplementary Materials for "Generative vs Discriminative: Rethinking The Meta-Continual Learning"

**Mohammadamin Banayeeanzade**[*], **Rasoul Mirzaiezadeh**[*],
**Hosein Hasani**[*], **Mahdieh Soleymani Baghshah**

Department of Computer Engineering
Sharif University of Technology

m.banayeean@gmail.com, mirzaierasoul75@gmail.com
hasanih@ce.sharif.edu, soleymani@sharif.edu

## 1 Justification of Statements

To justify the two statements mentioned in the paper, we consider a training dataset $\{(x_i, y_i)\}_{i=1}^{N}$, where each $x_i$ is generated from an underlying distribution determined by $y_i$, i.e. $x_i \sim \mathrm{P}^{y_i}(x \mid \theta^{y_i})$. $\theta^c$ denotes the unknown parameters for the distribution of class $c$. In the i.i.d. scenario, i.e. without presence of continual learning restrictions, a generative Bayes classifier would simply infer class-specific parameters either by solving maximum likelihood estimation ($\hat{\theta}^c = \underset{\theta^c}{\mathrm{argmax}}\,\mathrm{P}^c(\mathcal{S}^c \mid \theta^c)$) or by forming the posterior distribution ($P^c(\theta^c \mid \mathcal{S}^c)$). Notice that the mentioned formulas only depend on $\mathcal{S}^c$ and not the samples of other classes. This good property of generative models enables zero forgetting in the following continual scenarios:

**Statement 1.** In the *class-incremental scenario*, the data points for every class are delivered to the learner all at once. Hence, the parameter estimation or inference can be performed exactly as it was done in the i.i.d. setting. In fact, the inferred class-specific parameters would be the same in both continual and i.i.d. scenarios, resulting in the same decision boundaries.

**Statement 2.** In the *fully incremental scenario*, using incrementally-updatable sufficient statistics will guarantee zero forgetting. For more details, consider $T^c(x_1, ..., x_n)$ as the sufficient statistics for $P(\theta^c \mid x_1, ..., x_n)$ after observing $n$ samples of class $c$. A desired recursive form is $T^c(x_1, ..., x_n) = g(x_n, T^c(x_1, ..., x_{n-1}))$ where updating sufficient statistics of each class only depends on the last observed sample and the obtained sufficient statistic from previous samples. With this in mind, we see that observing the samples all at once will lead to the same sufficient statistics as the sequential case. Therefore inferring the posterior distribution will be the same for both continual and i.i.d. settings:

$$P^c(\theta^c \mid \mathcal{S}^c) = P^c(\theta^c \mid T^c(\mathcal{S}^c)) = P^c(\theta^c \mid T^c(x_1, ..., x_n)) = P^c(\theta^c \mid g(x_n, T^c(x_1, ..., x_{n-1}))). \tag{1}$$

Exponential family distributions, as a good example, have a closed-form predictive formulation and a convenient sufficient statistic:

$$P^c(x \mid \eta) \propto \exp(\boldsymbol{\eta}^c . \boldsymbol{T}^c(x)). \tag{2}$$

---

[*]Equal contribution

35th Conference on Neural Information Processing Systems (NeurIPS 2021).

The sufficient statistics for this family is:

$$\boldsymbol{T}^c(\mathcal{S}^c) = \sum_{x \in \mathcal{S}^c} \boldsymbol{T}^c(x), \tag{3}$$

with the incremental form easily written as:

$$\boldsymbol{T}^c(x_1, ..., x_n) = \boldsymbol{T}^c(x_1, ..., x_{n-1}) + \boldsymbol{T}^c(x_n). \tag{4}$$

## 2 Incremental Form for Gaussian Distributions

In the paper, we have mentioned that posterior parameters for class-specific Gaussian distributions can be written in an incremental form. In this section, we will investigate this claim in more detail.

Gaussian distributions are special cases of the exponential family, therefore as mentioned in the previous section, we will use the first and second-order moments as the incremental sufficient statistics:

$$\begin{aligned} \bar{z}_n &= \frac{1}{n} \sum_{i=1}^n z_i = \frac{(n-1)\bar{z}_{n-1} + z_n}{n} \\ \bar{\gamma}_n &= \frac{1}{n} \sum_{i=1}^n z_i^2 = \frac{(n-1)\bar{\gamma}_{n-1} + z_n}{n}, \end{aligned} \tag{5}$$

where $z_i$ is the embedded input in the $d$-dimensional space. With this in hand, we can write an incremental form for the posterior distribution parameters:

$$\kappa_n = \kappa_0 + n, \quad \mu_n = \frac{\kappa_0 \mu_0 + n\bar{z}_n}{\kappa_n},$$

$$\alpha_n = \alpha_0 + \frac{n}{2}, \quad \beta_n = \beta_0 + \frac{n}{2}(\bar{\gamma}_n - \bar{z}_n^2) + \frac{\kappa_0 n (\bar{z}_n - \mu_0)^2}{2(\kappa_0 + n)}, \tag{6}$$

where the class indicator $c$ and feature index $j$ are omitted for simplicity. It is easy to see that $\bar{z}_n \in \mathbb{R}^d$, $\bar{\gamma}_n \in \mathbb{R}_+^d$ and $n$ is a scalar. Therefore it suffices to store $2d + 1$ parameters for each class and update them as new samples for that class are arrived incrementally.

## 3 Hyperparameter Estimation

In this section, we first present an iterative algorithm for learning the prior distribution parameters. As mentioned in the paper, we are using the Normal-Gamma prior distribution as:

$$P\left(\mu^c, \lambda^c \mid \alpha_0, \beta_0, \kappa_0, \mu_0\right) \propto \prod_{j=1}^d \mathcal{N}\left(\mu_j^c \mid \mu_0, (\kappa_0 \lambda_j^c)^{-1}\right) Ga\left(\lambda_j^c \mid \alpha_0, \beta_0\right), \tag{7}$$

where $\mu^c$ and $\lambda^c$ are the mean and precision parameters for class $c$ in an $d$-dimensional embedding space, and $\alpha_0, \beta_0, \kappa_0$, and $\mu_0$ parameterize the prior distribution. We set $\kappa_0 = \mu_0 = 0$ to impose a flat uninformative prior on mean. However, the $\alpha_0, \beta_0$ are estimated using maximum likelihood on the inner-class precision of the features from the meta-train dataset.

$$\Lambda = \{\frac{1}{var(f_\psi(\mathcal{X}^c)_j)} \mid 0 \le j \le d, c \in \mathcal{C}_{\text{meta-train}}\}, \tag{8}$$

where $\mathcal{X}^c$ denotes all the data points from class $c$. Using these observations the likelihood function can be written as:

$$\begin{aligned} \log \mathcal{L}(\alpha, \beta; \Lambda) &= \sum_{\lambda \in \Lambda} \log Ga\left(\lambda \mid \alpha, \beta\right) = \sum_{\lambda \in \Lambda} \log \left(\frac{\beta^\alpha}{\Gamma(\alpha)} \lambda^{\alpha-1} e^{-\beta\lambda}\right) \\ &= n\alpha \log \beta - n \log \Gamma(\alpha) + (\alpha - 1) \sum_{\lambda \in \Lambda} \log \lambda - \sum_{\lambda \in \Lambda} \beta\lambda, \end{aligned} \tag{9}$$

where $n = | \Lambda |$. Applying straight forward maximization w.r.t. $\beta$ we get:

$$\beta = \frac{n\alpha}{\sum_{\lambda \in \Lambda} \lambda}, \tag{10}$$

plugging this into Equation 9, we get:

$$-n \log \Gamma(\alpha) + (\alpha - 1) \sum_{\lambda \in \Lambda} \log \lambda + n\alpha (\log n\alpha - \log \sum_{\lambda \in \Lambda} \lambda) - n\alpha. \tag{11}$$

Using the Taylor expansion, we can show:

$$\alpha \log \alpha \geq (1 + \log \alpha^0)(\alpha - \alpha^0) + \alpha^0 \log \alpha^0. \tag{12}$$

Substituting this into Equation 11, we get a lower bound on the likelihood function:

$$\log \mathcal{L}(\alpha; \Lambda) \geq -n \log \Gamma(\alpha) + (\alpha - 1) \sum_{\lambda \in \Lambda} \log \lambda - n\alpha$$
$$+ n((1 + \log \alpha^0)(\alpha - \alpha^0) + \alpha^0 \log \alpha^0) + n\alpha(\log n - \log \sum_{\lambda \in \Lambda} \lambda). \tag{13}$$

This lower bound is maximized at:

$$\alpha = \Psi^{-1}(\log n\alpha^0 + \sum_{\lambda \in \Lambda} \frac{\log \lambda}{n} - \log \sum_{\lambda \in \Lambda} \lambda), \tag{14}$$

where $\Psi$ denotes the digamma function. Given a good starting point for $\alpha^0$, we can use Equation 14 to solve the maximum likelihood. To find a good starting point, we use the Method of Moments:

$$\alpha^0 = \frac{\mu^2}{v},$$
$$\mu = \sum_{\lambda \in \Lambda} \lambda, \tag{15}$$
$$v = \frac{\sum_{\lambda \in \Lambda}(\lambda - \mu)^2}{n - 1}.$$

A combination of Equation 14 and Equation 15 produces Algorithm 1 for maximum likelihood estimation of Gamma distribution.

---

**Algorithm 1** Gamma Maximum Likelihood

---

**Require:** observation set $\Lambda$
1: $\mu = \sum_{\lambda \in \Lambda} \lambda$
2: $v = \frac{\sum_{\lambda \in \Lambda}(\lambda - \mu)^2}{n - 1}$
3: $\alpha = \frac{\mu^2}{v}$
4: **while** not converged **do**
5: $\quad \alpha = \Psi^{-1}(\log n\alpha + \sum_{\lambda \in \Lambda} \frac{\log \lambda}{n} - \log \sum_{\lambda \in \Lambda} \lambda)$
6: $\beta = \frac{n\alpha}{\sum_{\lambda \in \Lambda} \lambda}$
7: **return** $\alpha, \beta$

---

### 3.1 Maximum Likelihood in Practice

Figure 1 shows that a proportion of features are inactive in each class, meaning that they have zero (or close to zero) inner-class variance. The inactive features vary for different classes. As shown in Figure 1 as we increase the embedding size by simply changing the input size from $28 \times 28$ to $84 \times 84$ in Omniglot, the proportion of inactive features increases. Moreover, with the same input and embedding size, this proportion also increases as the dataset complexity decreases from Mini-ImageNet to Omniglot ($84 \times 84$). These inactive features cause instability in the maximum likelihood estimation as the precision approaches infinity. Moreover, an increase in the number of inactive features causes a heavy bias towards zero in the prior distribution, this bias results in unstable zero-variance posterior. To make our method robust to dataset complexity and embedding size changes, we propose three approaches and discuss their results.

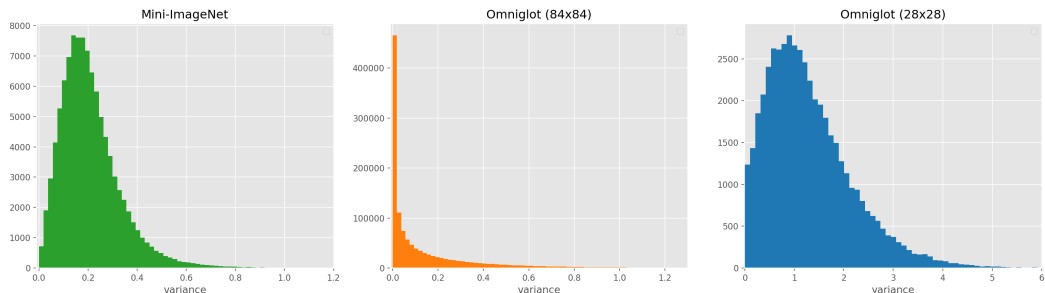

Figure 1: Distribution of intra-class variances on the embedding features from Mini-ImageNet and Omniglot meta-train datasets.

Table 1: Effect of hyperparameter estimation procedure on the performance of the GeMCL method.

| Method
Dataset | Naïve | Mean of
Variances | Filtering
Variances | Parameter
Scaling |
|---|---|---|---|---|
| Mini-ImageNet | $57.0_{\pm 0.9}$ | $59.2_{\pm 0.9}$ | $59.1_{\pm 0.9}$ | $59.2_{\pm 1.0}$ |
| Omniglot | $68.8_{\pm 0.7}$ | $86.9_{\pm 0.6}$ | $86.8_{\pm 0.6}$ | $86.8_{\pm 0.6}$ |
| Omniglot ($28 \times 28$) | $83.2_{\pm 0.7}$ | $84.4_{\pm 0.6}$ | $84.4_{\pm 0.6}$ | $84.1_{\pm 0.6}$ |

**Mean of Variances:** In order to reduce the effect of too high (noisy feature) or too low (inactive feature) variances, we use a summary of feature variances for each class instead of using each feature variance. In other words, the observation set is changed to:

$$\Lambda = \{ \frac{d}{\sum_{0 \leq j \leq d} var(f_\psi(\mathcal{X}^c)_j)} \mid c \in \mathcal{C}_{\text{meta-train}} \}, \tag{16}$$

this change reduces the size of $\Lambda$ but produces more stable samples, resulting in a better prior which is not heavily biased towards inactive or noisy features.

**Filtering Variances:** Another way of resolving the inactive and noisy variances is by omitting $l$ smallest and $h$ largest variances. Obviously, this method reduces the noise and unwanted bias caused by the noisy or inactive features, by selecting a set of more stable features for each class. As the behavior of features varies based on the dataset and architecture, we employ a search among different values for $h$ and $l$ to select the best-performing ones based on the meta-validation set.

**Parameter Scaling:** Unlike the previous approaches which modify the observation set $\Lambda$, in this approach, we modify the prior distribution directly. After the maximum likelihood estimation on the observation set, we scale both $\alpha$ and $\beta$ parameters by a factor $C$. This modification preserves the mean value of the distribution while changing its mode to reduce its bias towards zero. Same as the filtering approach this constant factor needs to be tuned based on the meta-validation set.

In order to evaluate these approaches, we perform experiments on Mini-ImageNet and Omniglot datasets as shown in Table 1. The acceptable performance of the naive approach in Mini-ImageNet and Omniglot ($28 \times 28$), which is in line with the lower bias shown in Figure 1, show that it performs well when the embedding dimension matches the dataset complexity. However, the robustness and better performance of the proposed approaches result in a method, applicable to all the settings without worrying about the embedding size. Among the proposed methods using the mean of variances shows slightly better performance without the need for further hyperparameter tuning, therefore we select this approach for other experiments.

# 4 GeMCL Algorithm

As mentioned in the paper, our meta-training algorithm will first optimize embedding parameters based on the following loss function:

$$\mathcal{L}(\psi) = -\sum_{(x,y)\in\mathcal{Q}} \log P\left(y \mid f_\psi(x), f_\psi(\mathcal{S}_X), \mathcal{S}_Y, \alpha_0, \beta_0\right)$$

$$= -\sum_{(x,y)\in\mathcal{Q}} \log \frac{P(f_\psi(x)|y, f_\psi(\mathcal{S}_X), \mathcal{S}_Y, \alpha_0, \beta_0)}{\sum_{\hat{y}=1}^{C} P(f_\psi(x)|\hat{y}, f_\psi(\mathcal{S}_X), \mathcal{S}_Y, \alpha_0, \beta_0)}$$

$$= -\sum_{(x,y)\in\mathcal{Q}} \log \frac{P\left(f_\psi(x)|f_\psi(\mathcal{S}_X^y), \alpha_0, \beta_0\right)}{\sum_{\hat{y}=1}^{C} P\left(f_\psi(x)|f_\psi(\mathcal{S}_X^{\hat{y}}), \alpha_0, \beta_0\right)} \tag{17}$$

$$= -\sum_{(x,y)\in\mathcal{Q}} \log \text{softmax}\left(\log P\left(f_\psi(x) \mid f_\psi(\mathcal{S}_X^y), \alpha_0, \beta_0\right)\right).$$

In the above formulations, we assumed a uniform distribution on class labels for simplicity in notation. It is easy to generalize to the case of unbalanced class distributions. The $\alpha_0$ and $\beta_0$ are kept constant while training the feature extractor with the values of 100 and 1000 respectively. When the feature extractor is trained, the $\alpha_0$ and $\beta_0$ are estimated using maximum likelihood, as discussed in Section 3.

The detailed steps of meta-train and meta-test phases are described in Algorithm 2 and Algorithm 3 respectively.

---

**Algorithm 2** GeMCL meta-train phase

---

**Require:** set of continual learning problems, i.e. $\mathcal{D}_{meta-train}$
 1: initialize $\alpha_0, \beta_0$ with large values
 2: randomly initialize $\psi$
 3: **while** not done **do**
 4:     $\mathcal{M} = \{\}$                                                           ▷ class-specific parameters
 5:     sample a continual problem $(\mathcal{S}, \mathcal{Q})$ from $\mathcal{D}_{meta-train}$
 6:     **for** $\mathcal{S}_t \in \mathcal{S}$ **do**
 7:         **for** $(x_i, y_i) \in \mathcal{S}_t$ **do**
 8:             **if** $y_i \notin \mathcal{M}$ **then**
 9:                 $\mathcal{M}[y_i] \leftarrow (0, 0, 0)$                      ▷ initialize parameters for class $y_i$
10:             $(n^{y_i}, \bar{z}_n^{y_i}, \bar{\gamma}_n^{y_i}) \leftarrow \mathcal{M}[y_i]$
11:             $z_i = f_\psi(x_i)$
12:             update the first and second-order moments (Eq. 5)
13:             $\mathcal{M}[y_i] \leftarrow (n^{y_i} + 1, \bar{z}_{n+1}^{y_i}, \bar{\gamma}_{n+1}^{y_i})$
14:     compute posterior parameters for all classes (Eq. 6)
15:     optimize $\mathcal{L}(\psi)$ with respect to $\psi$ over samples in $\mathcal{Q}$ (Eq. 17)
16: freeze $\psi$
17: compute $\Lambda$ from $\mathcal{D}_{meta-train}$ (Eq. 16)
18: compute $\alpha_0$ and $\beta_0$ using Alg. 1 and $\Lambda$
19: **return** $\alpha_0, \beta_0, f_\psi$

---

# 5 Backbones Comparison

As mentioned in the paper, we incorporate a well-studied backbone for our experiments which uses less than 115K parameters [2]. We did not explore further modifications of the architecture to optimize it for our proposed method. OML has introduced a backbone that uses more than 2.9M parameters and is designed to fit their method. Although their backbone imposes feature sparsity and also dead-neurons in our case, in Table 2 we have included the result of the GeMCL method on the OML backbone. Moreover, adding a batch normalization after each layer of the OML backbone significantly reduces the number of dead neurons and increases the accuracy. Note that, as the ANML utilizes a method-specific architecture with a mask branch, we did not include it in this experiment.

**Algorithm 3** GeMCL meta-test phase

---

**Require:** a continual problem : $(\mathcal{S}, \mathcal{Q})$
**Require:** $f_\psi, \alpha_0, \beta_0$

1: $\mathcal{M} = \{\}$                          ▷ class-specific parameters
2: **for** $\mathcal{S}_t \in \mathcal{S}$ **do**
3:      **for** $(x_i, y_i) \in \mathcal{S}_t$ **do**
4:          **if** $y_i \notin \mathcal{M}$ **then**
5:              $\mathcal{M}[y_i] \leftarrow (0, 0, 0)$            ▷ initialize parameters for class $y_i$
6:          $(n^{y_i}, \bar{z}_n^{y_i}, \bar{\gamma}_n^{y_i}) \leftarrow \mathcal{M}[y_i]$
7:          $z_i = f_\psi(x_i)$
8:          update the first and second-order moments (Eq. 5)
9:          $\mathcal{M}[y_i] \leftarrow (n^{y_i} + 1, \bar{z}_{n+1}^{y_i}, \bar{\gamma}_{n+1}^{y_i})$
10: compute posterior parameters for all classes (Eq. 6)
11: **for** $x_i \in \mathcal{Q}$ **do**
12:      $\hat{y}_i = \underset{y}{\arg\max} P\left(f_\psi(x_i) \mid \alpha_n^y, \beta_n^y, \mu_n^y, \kappa_n^y\right)$

---

Table 2: Performance of the GeMCL model using OML backbone along with the standard performance of methods.

| Method \ Dataset | Mini-ImageNet | Omniglot |
|---|---|---|
| OML | $22.1_{\pm1.4}$ | $21.1_{\pm0.7}$ |
| GeMCL (OML Backbone) | $47.3_{\pm1.0}$ | $76.6_{\pm0.7}$ |
| GeMCL (OML Backbone + BN) | $59.7_{\pm1.0}$ | $85.0_{\pm0.8}$ |
| GeMCL | $59.2_{\pm0.9}$ | $86.9_{\pm0.6}$ |

## 6   Study of the Feature Extractors

All of the mentioned meta-continual methods use meta-train data to provide a rich feature extractor and then perform further adaptation to meta-test data. Using the same method for meta-training and meta-testing is reasonable, since each method may perform poorly on the networks meta-trained with other methods. However, analyzing each method on feature extractors produced by various methods can provide more insights into the power and generality of that method. Moreover, it also shows the strength of the feature extractors provided by each method. As shown in Tables 3 and 4, the GeMCL variants outperform other methods on various feature extractors. The feature extractor provided by the GeMCL and PGLR methods have also high accuracy across different models.

Table 3: Performance of the proposed baselines on the Mini-ImageNet dataset using feature extractors trained by various methods. The blue highlighted cells indicate the performance of each model on its standard feature extractor. The last row shows the average accuracy of the corresponding method across feature extractors.

| Network \ Test Method | MTLR | PGLR | Prototypical | GeMCL-MAP | GeMCL |
|---|---|---|---|---|---|
| Random | $5.5_{\pm0.9}$ | $15.3_{\pm1.2}$ | $17.8_{\pm0.8}$ | $18.9_{\pm0.8}$ | $18.9_{\pm0.8}$ |
| Pretrained | $32.8_{\pm3.1}$ | $54.2_{\pm1.0}$ | $55.6_{\pm1.1}$ | $55.9_{\pm1.1}$ | $55.9_{\pm1.1}$ |
| MTLR | $31.1_{\pm1.1}$ | $43.4_{\pm1.1}$ | $43.2_{\pm1.0}$ | $43.3_{\pm0.9}$ | $43.4_{\pm0.9}$ |
| PGLR | $19.6_{\pm3.9}$ | $53.2_{\pm1.1}$ | $56.8_{\pm1.0}$ | $57.0_{\pm1.1}$ | $57.0_{\pm1.1}$ |
| Prototypical | $5.0_{\pm0.0}$ | $46.7_{\pm1.4}$ | $54.8_{\pm0.9}$ | $55.4_{\pm0.9}$ | $55.4_{\pm1.0}$ |
| GeMCL-MAP | $23.6_{\pm4.2}$ | $56.3_{\pm1.0}$ | $58.4_{\pm1.0}$ | $58.7_{\pm1.0}$ | $58.7_{\pm1.0}$ |
| GeMCL | $26.0_{\pm4.0}$ | $56.6_{\pm1.1}$ | $58.7_{\pm1.0}$ | $59.2_{\pm0.9}$ | $59.2_{\pm0.9}$ |
| Average | 20.5 | 46.5 | 49.3 | 49.8 | 49.8 |

Table 4: Performance of the proposed baselines on the Omniglot dataset using feature extractors trained by various methods.

| Network \ Test Method | MTLR | PGLR | Prototypical | GeMCL-MAP | GeMCL |
|---|---|---|---|---|---|
| Random | $4.4_{\pm 0.1}$ | $32.7_{\pm 0.8}$ | $36.2_{\pm 0.8}$ | $38.1_{\pm 0.8}$ | $37.6_{\pm 0.8}$ |
| Pretrained | $23.6_{\pm 0.7}$ | $61.0_{\pm 0.8}$ | $73.9_{\pm 0.6}$ | $76.3_{\pm 0.8}$ | $75.9_{\pm 0.8}$ |
| MTLR | $61.9_{\pm 0.7}$ | $74.6_{\pm 0.6}$ | $82.0_{\pm 0.6}$ | $84.3_{\pm 0.7}$ | $84.9_{\pm 0.6}$ |
| PGLR | $39.1_{\pm 0.7}$ | $72.8_{\pm 0.6}$ | $87.0_{\pm 0.5}$ | $87.9_{\pm 0.5}$ | $87.4_{\pm 0.5}$ |
| Prototypical | $1.9_{\pm 0.3}$ | $79.6_{\pm 0.7}$ | $83.7_{\pm 0.6}$ | $83.6_{\pm 0.5}$ | $83.7_{\pm 0.5}$ |
| GeMCL-MAP | $20.0_{\pm 1.8}$ | $82.7_{\pm 0.6}$ | $86.4_{\pm 0.5}$ | $86.9_{\pm 0.6}$ | $86.7_{\pm 0.6}$ |
| GeMCL | $19.8_{\pm 1.9}$ | $82.8_{\pm 0.6}$ | $86.6_{\pm 0.6}$ | $87.0_{\pm 0.6}$ | $86.9_{\pm 0.6}$ |
| Average | 25.0 | 70.1 | 76.5 | 77.7 | 77.6 |

# 7 Effect of Learning Rate

The inner learning rate of the discriminative baselines is the most important factor that affects the catastrophic forgetting analysis. High values of learning rate may result in higher accuracy for the current task, but it may also increase the forgetting of the earlier tasks. On the other hand, small values for the learning rate may reduce the catastrophic forgetting, but may also reduce the performance on newer tasks. This issue relates to the so-called stability-plasticity dilemma [1]. Keeping the learning rate fixed, the number of training epochs imposes the same dilemma.

In the paper, we use the plots of the first-class accuracy to analyze the forgetting issue, and plots of the average accuracy to analyze the ability of adaptation and generalization capability of baselines.

Learning rate tuning for discriminative baselines is necessary to find the best setting. OML and ANML perform a learning rate search on the meta-test data and find the best learning rate for the final experiment. Even for a minor change in experiment setup, e.g. changing the number of ways or shots, they retune the learning rate. In fact, the abnormal changes of the OML performance, like the increase in the 450-way performance in Figure 4 of the paper, are related to this issue.

Since this use of meta-test data is counter-intuitive, we tune the learning rate of our MTLR and PGLR baselines using meta-validation data once the feature extractor is trained and use it across different numbers of shots and ways. We also retune the learning rate of MTLR for i.i.d settings in Table 3 of the paper, since the optimal learning rate for the continual setting is too small for the i.i.d. setting. However, it is more desirable if the optimal learning rate for the i.i.d. setting of a baseline performs well on the continual setting too. Hence in Table 5, we analyze the performance of the baselines in a fair scenario, in which the performance of the MTLR and PGLR models are compared in the same situations. As shown, substantial gaps between the performance of MTLR in different tuning settings show that this model is extremely sensitive to the learning tuning setting. As opposed to the MTLR model, the PGLR model is more stable across the different learning rate tuning schemes and it is robust to forgetting, even when the learning rate is tuned for the i.i.d. setting. It is worth mentioning that the GeMCL model does not require learning rate tuning and also performs identically in the continual and i.i.d. scenarios.

# 8 MTLR and PGLR Baselines

MTLR and PGLR are two discriminative methods based on logistic regression that both utilize optimization-based meta-learning to achieve a proper feature extractor. The classification head of the MTLR model is a simple logistic regression that is updated via gradient descend. It mimics the idea from the OML model but removes additional details such as 1: Maintaining a classifier head with an output for each class across all meta-train episodes, 2: Using out-of-episode query samples in meta-training, 3: Using a two-layer classifier head, 4: Visiting the data of meta-test episodes multiple times to tune inner-learning rate. The PGLR model possesses the following modifications compared with the standard logistic regression:

- Normalization of the class-related weights

Table 5: Effect of learning tuning procedure on the performance of selected baselines.

| Dataset | Learning Rate Tuning Setting | Mode | Head for Pretrained Backbone | | | Meta-train | | |
|---|---|---|---|---|---|---|---|---|
| | | | MTLR | PGLR | GeMCL | MTLR | PGLR | GeMCL |
| Mini-ImageNet | Continual | Continual | $32.8_{\pm 3.1}$ | $54.2_{\pm 1.0}$ | $55.9_{\pm 1.1}$ | $31.1_{\pm 1.1}$ | $53.2_{\pm 1.1}$ | $59.2_{\pm 0.9}$ |
| | | IID | $38.5_{\pm 1.4}$ | $55.8_{\pm 1.0}$ | $55.9_{\pm 1.1}$ | $31.5_{\pm 1.1}$ | $57.0_{\pm 0.9}$ | $59.2_{\pm 0.9}$ |
| | IID | Continual | $7.1_{\pm 1.9}$ | $53.3_{\pm 1.0}$ | $55.9_{\pm 1.1}$ | $22.3_{\pm 1.4}$ | $51.4_{\pm 1.4}$ | $59.2_{\pm 0.9}$ |
| | | IID | $60.9_{\pm 1.0}$ | $56.2_{\pm 1.0}$ | $55.9_{\pm 1.1}$ | $48.9_{\pm 1.1}$ | $57.4_{\pm 1.0}$ | $59.2_{\pm 0.9}$ |
| Omniglot | Continual | Continual | $23.6_{\pm 0.7}$ | $61.0_{\pm 0.8}$ | $75.9_{\pm 0.8}$ | $61.9_{\pm 0.7}$ | $72.8_{\pm 0.6}$ | $86.9_{\pm 0.6}$ |
| | | IID | $40.5_{\pm 0.8}$ | $74.2_{\pm 0.6}$ | $75.9_{\pm 0.8}$ | $74.5_{\pm 0.6}$ | $86.8_{\pm 0.5}$ | $86.9_{\pm 0.6}$ |
| | IID | Continual | $4.9_{\pm 1.0}$ | $68.3_{\pm 0.7}$ | $75.9_{\pm 0.8}$ | $46.6_{\pm 1.1}$ | $79.1_{\pm 0.6}$ | $86.9_{\pm 0.6}$ |
| | | IID | $77.8_{\pm 0.7}$ | $74.6_{\pm 0.8}$ | $75.9_{\pm 0.8}$ | $82.9_{\pm 0.5}$ | $86.8_{\pm 0.5}$ | $86.9_{\pm 0.6}$ |

- Performing gradient descent only through the class-related weights

Here we aim to analyze the effectiveness of the aforementioned components. We introduce two new variants for the PGLR model by removing each one of these components. The PGLR/Norm is a PGLR model without the first component, and the PGLR/Opt is a PGLR model without the second one.

To evaluate the methods impartially, here we remove the need for feature extractors, by comparing the performance of classifier heads on random 10-dimensional datasets. For better comparison, extra conditions including the learning rate and the number of epochs should also be identical. However, as discussed in Section 7, different models have different stability-plasticity criteria, and choosing the same setting for different models is not fair. Hence, we propose a new criterion for comparison. We choose a proper but fixed learning rate for all of the models but the number of training epochs of each task is different. Each model continues learning on a specific task until its performance on validation data of that task reaches a minimum threshold. In this scenario, the amount of learning (plasticity) in different models is controlled to be relatively identical. So, with the same amount of learning, the comparison of robustness to forgetting (stability) becomes more impartial.

Figure 2 shows the performance of the PGLR model and its variants along the MTLR and generative approaches on the suggested toy datasets. As shown, both of the PGLR components are effective in reducing forgetting. Moreover, when combined in the PGLR model, they offer considerable robustness to the forgetting. Note that the described stability-plasticity dilemma and controlling the learning rate effect is not an issue in the generative approaches and reduction in their performance is due to the generalization issue.

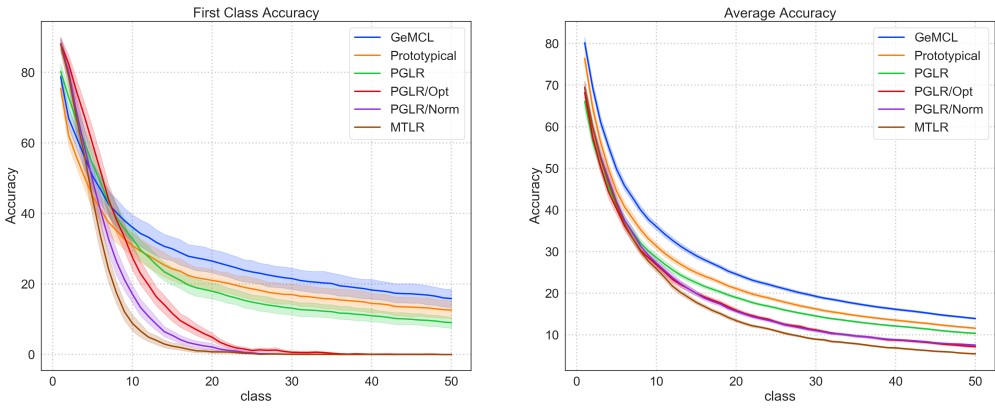

Figure 2: Performance of classifier heads on 10-dimensional toy datasets. The pale margins indicate confidence intervals across different datasets.

Table 6: Approximate running time of the proposed baselines on the GTX 1080 Ti GPU.

| Dataset | Mode | # | | | | Method | | | |
| | | Episodes | Ways | Shots | Queries | MTLR | PGLR | Prototypical | GeMCL |
|---|---|---|---|---|---|---|---|---|---|
| Mini-ImageNet | Train | 30K | 20 | 10 | 30 | 170 min | 200 min | 130 min | 140 min |
| | Test | 100 | 20 | 100 | 100 | 70 sec | 120 sec | 10 sec | 20 sec |
| Omniglot | Train | 20K | 20 | 5 | 15 | 110 min | 130 min | 90 min | 90 min |
| | Test | 100 | 600 | 15 | 5 | 320 sec | 620 sec | 120 sec | 240 sec |

## 9 Computing Infrastructure and Further Experiment Details

We have performed our experiments on the two separate devices with GeForce GTX 1080 Ti GPU. Since the main approach uses a light backbone and the training procedure does not use second-order optimization, the overall running time is low. Table 6 compares the duration of training and test time of different methods on the same device. It shows our method can be used with much less energy consumption while achieving better results.

To pretrain the backbone on datasets we put the meta-train and meta-validation classes together to be used both in training and validation. The validation split is constructed by taking $20\%$ of data for each class and the remaining data is used for training. The backbone is pretrained for 1000 epochs with a batch size of 64 on Mini-ImageNet, and 50 epochs with batches of size 32 for Omniglot.

Throughout this paper, all experiments reported with an error bar have been repeated in 100 random test episodes. To make results standard and comparable, all of the tables that include experiments on the Mini-ImageNet dataset are performed with 20-way 100-shot settings, and Omniglot experiments are performed with 600-way 15-shot settings. These settings are chosen to better reflect the performance of models in continual learning scenarios, however, other settings also result in similar outcomes preserving relative performance between models (See Figure 4 and 5 in the paper).

## 10 Code and Data Availability

Related source code is available at `https://github.com/aminbana/GeMCL`. In this repository, there exists a README file containing instructions, configuration details, and path to the trained models. Moreover, the licenses of the freely available datasets and used source codes are also available in the README file.