# OpenReview forum: "Generative vs. Discriminative: Rethinking The Meta-Continual Learning"
_NeurIPS.cc/2021/Conference — NeurIPS 2021 Poster_

### Official Review · Reviewer_TJWB · 2021-07-10

**Rating:** 7
**Confidence:** 3

**Summary:**

The authors propose the generative classifier for meta continual learning. This approach is an extension of a well-known method for few-shot learning called prototypical networks. The proposed method is evaluated on two challenging continual learning datasets (Omniglot and MiniImagenet) and compared to other methodsin this field as well as adequate baselines.
The main contributions of the paper can be summarized as:
- The introduction of the new method for continual learning using the generative classifier.
- Clearly explained and visualized with toy example the difference between catastrophic forgetting in case of generative and discriminative classifiers.
- Precisely formulated and justified statements about immunity to catastrophic forgetting.
- Insightful empirical analysis of the proposed method (mostly covered in the appendix).


**Limitations And Societal Impact:**

The authors correctly identified the main limitations of this method, however, did not provide any discussion on how these limitations can be circumvented. Additionally, the authors did not provide any discussion about how this method can be further developed and what are the possible future research paths. The other issue is the evaluation procedure for the method. To assign class for any datasample during the test the method finds maximum probability over all already known class distributions. This leads to the linear growth of the method runtime and memory requirements  with respect to the classes. I would be interested to see if the authors has any potential ideas how to reduce this requirements?

**Main Review:**

Originality: The paper presents an approach to meta continual learning using a generative classifier. The proposed method is closely related to the method introduced for few-shot learning called Prototypical Networks. Authors points out that Prototypical Networks is a special case of their method. This suggests that the proposed method should be traeted as a valuable extension of the existing algorithm rather than a novel approach.
The proposed method is immune to catastrophic forgetting i.e., at the end of continual training, its performance is equal to the performance of the same model in standard i.i.d setting, which is remarkable. However, this is achieved explicitly by two design choices: static embedding network, which is frozen after the meta training phase, and parameter isolation enforced by the generative classifier. While this approach offers immunity to catastrophic forgetting, it simultaneously forbids the network to continuously adapt to upcoming data and reuse previously learned knowledge with transfer learning. This raises the question of whether the approach is a continual learning method.

Quality: The paper is a complete piece of work that starts with introducing the general idea backed up by the toy example demonstrating the advantage of the proposed solution. Next section presents the mathematical details to conclude the paper with the section of empirical results validating the proposed method. The experimental section is thorough and well written and the proposed baselines are sensible. However, I have a few questions which were not answered in this section. I will be more than happy to discuss them with the authors. The statements presented in the paper are precise, and the argumentation in favor of these statements is convincing. The only exception is the claim that the authors make in the introduction section: “Encouraged by generative classifiers, we propose a new update scheme for discriminative classifiers to reduce the catastrophic forgetting problem.” If I understand correctly, the authors refer here to the PGLR baseline in which only the part of the classifying layer is updated according to the present task. I find it challenging to understand the novelty of this method since all the classifiers working in the regime of Task Incremental scenario works the same way. I would appreciate it if the authors elaborate on this topic.

Additional questions to the authors:
The model’s effectiveness is linked to the quality of the pretrained embedding network, and the paper lacks the thorough discussion of this part. I enlist the following questions, which may help to clarify this part.
In all experiments the meta train split forms a larger part of the whole dataset. How the size of the meta-train dataset impacts the overall performance of the method? Does the perforamcen plateaus at some point or does it constantly improve with the bigger meta train dataset?
Regarding results in Table 2 (appnedix). Why the use of OML backbone results in worse results? This backbone is almost 40 x bigger than the backbone used in the main experiments, yet it achieves significantly worse results. Is it a general tendency that the bigger the backbone the worse the results?
I miss the details about learning procedure of the pretrained backbone. In Table 1 (main paper) the difference between pretrained backbone and meta learning on ImageNet is relatively small while in case of Omniglot is substantially bigger. What might be the reason for this?

The last question is rather technical. In Fig.4 (main paper) X axis ends on 600th class while the test set should be composed of 723 classes because the meta train and meta validation split has 900 classes and the whole dataset is composed of 1623. What is the reason for this?


Clarity: This paper is well written and has a correct structure. The code enclosed in supplementary files is well-structured and easy to follow. Additionally, the details for experimental results are clearly presented.

Significance: Authors presents the method clearly and evaluates is effectiveness on challenging benchmarks on which it achieves state-of-the-art results compared to other methods in this field. However, as mentioned earlier, I find it inappropriate to classify this method as a continual learning method due to its abovementioned limitations. Nevertheless, the proposed solution to use the generative approach for classification in continual learning is a valuable insight. It could serve as a building block for other methods in this field.

**Time Spent Reviewing:**

6

---

> ### Author Response · Authors · 2021-08-09
> **Response to reviewer TJWB**
>
> First of all, we would like to thank you for the thorough review of the paper and for your insightful questions.
>
> --**This raises the question of whether the approach is a continual learning method.**
>
>
> To the best of our knowledge, the main purpose of continual learning is to alleviate catastrophic forgetting and the positive forward/backward transfers are additional desired properties. For example, the authors in [1] use a fixed random network for all tasks and learn a binary mask over it for each task separately. The acquired mask is frozen permanently and is later used in the testing phase. It is clear that there is no transfer across tasks. Furthermore, the investigations in [3] suggest that the early representation layers remain almost unchanged even if we allow them to adapt to new incoming tasks. In fact they emphasize that classification heads are the most responsible parts for forgetting (refer to our paper lines 169 - 173).
>
> The OML suffers from both forgetting (by using a discriminative classifier) and almost no positive knowledge transfer (by freezing the backbone parameters). We have mainly focused on solving the former problem by using generative classifiers. ّIn the future work, we hope that using a Bayesian perspective for updating the backbone weights (e.g. using VCL method[2]) combined with a meta-learning trick may lead to better transfer.
>
> Talking about the future research path, we also propose updating the class-related parameters in the embedding space before answering the query samples, e.g. by passing them through a set-to-set function as proposed in FEAT for few-shot classification. Updating the class-related parameters enables knowledge transfer between different tasks seen during an episode. Moreover, similar to some continual learning methods, a replay memory can also be used to help the adaptation process of these parameters.
>
>
> --**I find it challenging to understand the novelty of this method since all the classifiers working in the regime of Task Incremental scenario works the same way**
>
>
> The PGLR method has two components (lines 240-243 main paper and 148-149 of the supplementary materials) that together make the model more robust to forgetting while producing remarkable accuracy compared to a simple discriminative baseline (MTLR). The effect of each of these components is studied in supplementary materials (Section 8 and Figure 3). We can see that excluding each of these components from the PGLR model, significantly reduces robustness to forgetting. Interestingly, by analyzing these components solely, class-related normalization shows more effectiveness than updating class-related weights. One possible reason behind this phenomenon is that updating weights without class normalization may result in imbalanced class weights (comparing the first seen classes and the last ones), even when irrelevant weights have not been updated during training.  Hence, we believe that the PGLR method is fairly novel as a whole (with both components) in the literature. However, the mentioned sentence will be edited to make it clear that we are talking about the use of both the explained components.
>
>
> --**How the size of the meta-train dataset impacts the overall performance of the method?**
>
> | Meta-train size		 | 25%	 | 50%	 	| 75%     | 100%	 |
> | ---------------------- | ----- | -------- | ------- | -------- |
> | Omniglot Acc.	 		 | 82.5	 | 85.6	 	| 86.6    | 86.9	 |
> | Mini-Imagenet Acc.	 | 44.1	 | 56.0     | 58.0    | 59.2	 |
>
> Compared to the Mini-Imagenet dataset, the rate of improvement in the Omniglot dataset is lower. This is probably because visual features in the Omniglot dataset are more simple and also similar across different classes.
>
> --**Is it a general tendency that the bigger the backbone the worse the results?**
>
> The reason behind this reduction in performance is not the size of backbone, in fact, it is the design of the backbone! As mentioned in the supplementary materials (lines 101-104), the OML backbone seems to be designed to increase sparsity. Moreover, we observed that it extensively imposes dead-neurons (neurons that are inactive for every input) in the embeddings. To support our claim we simply added a batch normalization after each layer of the OML backbone. This simple modification resulted in $12.4\\%$, $8.4\\%$ performance increase in Mini-Imagenet and Omniglot datasets respectively.
>
> --**I miss the details about learning procedure of the pretrained backbone**
>
> The pretraining details (learning rate, train classes, epochs, batch size, ...) are left to the parameters submitted with source codes, and the overall procedure is straightforward. We will also produce details in the supplementary materials for readers.
>
> --**In Table 1 (main paper) the difference between pretrained backbone and meta learning on ImageNet is relatively small while in case of Omniglot is substantially bigger. What might be the reason for this?**
> Our hypothesis is that this relates to the high number of train classes in Omniglot. As in pretraining (batch learning) at each step, we sample a relatively small batch to update all the parameters. Obviously, this batch does not contain samples for all classes, in fact, it is a small portion compared to the number of classes in Omniglot. This causes inaccurate classification heads throughout training and these classification heads indeed affect the learning of the feature extractor parameters. Note that this is not the case for meta-learning where we only have classification heads for the present classes in each episode. Another reason could be the few-shot nature of Omniglot, as in each meta-training episode we always have all 20 samples of the present classes which results in a fairly accurate loss function. However, in the Mini-Imagenet dataset, a relatively small portion of class samples appears in each episode.
>
>
>
> --**The last question is rather technical. In Fig.4 (main paper) X axis ends on 600th class while the test set should be composed of 723 classes …**
>
> In fact, there exists a typo in the dataset description section. Actually, there are 963 classes for meta-training (763 for training and 200 for validation) based on the Omniglot standard split[4]. This leaves 660 classes for meta-testing. We have followed the experiment setup of ANML, which has used 600 classes for the test. However, the code provided in the supplementary can be easily adjusted and there is no problem for testing with 660 classes. Accordingly the average accuracy for 660 classes is $86.1\\%$ compared to $86.9\\%$ for 600 classes.
>
>
> --**Additionally, the authors did not provide any discussion about how this method can be further developed and what are the possible future research paths**
>
> Thanks for your constructive comment, we will add ideas for future works to the paper. As we explained earlier in this comment, some future directions could be: (a) Applying set-to-set functions or optimization to enhance the class-related parameters and enable inter-task knowledge transfer. (b)  Using a Bayesian perspective (introduced by VCL[2]) for updating the backbone weights during the meta-test phase.
>
> --**This leads to the linear growth of the method runtime and memory requirements with respect to the classes. I would be interested to see if the authors has any potential ideas how to reduce this requirements?**
>
> This concern is somehow addressed in the main paper (in lines 189 - 193). In fact, the linear growth problem is an innate issue for all continual learning methods. In the “class incremental” scenario where the final classification is done across all observed classes, every continual learning method has the obligation to expand the weights of its classifier linearly with respect to the number of observed classes. Even considering a simple linear classifier as the naivest possible continual learner, the corresponding weights of this classifier has to extend with the number of streamed classes to support the multi-class classification. Furthermore, our memory growth is extremely small, because we merely store low-dimensional parameters (line 34 of supplementary materials) in the embedding space (rather than raw high-dimensional input samples, new backbone weights, network masks, or etc that have been used in many related methods) which is an inevitable requirement for every possible continual classifier.
>
> Besides the zero forgetting property of the generative classifiers, the strength of our method is that the class-related parameters can be constructed in an on-demand online fashion, i.e. by observing the samples for a new class, we can easily form its parameters without making any change to the other class parameters. We are not restricted to an upper bound on the number of incoming classes. This is in contrast with OML, who uses a fixed upper bound.
>
>
>
> [1] Supermasks in Superposition, NeurIPS 2020, https://proceedings.neurips.cc/paper/2020/hash/ad1f8bb9b51f023cdc80cf94bb615aa9-Abstract.html
>
>
> [2] Variational Continual Learning,  https://arxiv.org/abs/1710.10628
>
>
> [3]Anatomy of Catastrophic Forgetting: Hidden Representations and Task Semantics, https://arxiv.org/abs/2007.07400
>
> [4] https://github.com/brendenlake/omniglot

---

> > ### Comment · Reviewer_TJWB · 2021-08-18
> > **Rebuttal response**
> >
> > Dear authors. Thank you for submitting the rebuttal and answering my questions. After careful read I increase my score to 7.

---

### Official Review · Reviewer_DcGp · 2021-07-17

**Rating:** 6
**Confidence:** 4

**Summary:**

This paper proposes a probabilistic generative approach using meta-learning for continual learning. It is inspired by cognitive and neuroscience about meta-continual learning to address catastrophic forgetting. It presents the problems occurring in the discriminative approaches and suggests how to handle them by using generative classifiers. Using new update scheme for discriminative
classifiers, it shows that the catastrophic forgetting problem is reduced.
It also improves the existing prototypical learning by Bayesian approach.





**Limitations And Societal Impact:**

Please see the weaknesses in the main review section.  Include the block diagram for the proposed approach in the main paper.

**Main Review:**

Strong Points:
1- The paper is well organized and easy to follow. It provides sufficient information for each component used in the model and sare code for cross-checking the results.

2-  It shows the use of meta-learning in continual learning and presents the advantage of the generative model over the discriminative model. It proposes a way to improve the performance of the discriminative model by a new updated scheme that reduces the catastrophic forgetting problem.

3-  The proposed model is implemented in the fully incremental
setting. It includes a solid ablation analysis for each component.

Weaknesses:
1-  There are several existing approaches for Meta-Continual Learning [a,b].  It is unclear to me the exact differences between the existing and the proposed method except for the generative approach.  Would you please provide a precise difference and justify how it handles the shortcoming compared to previous approaches.

2- Experimental results are missing for the CIFR-100 dataset.

3- Can you apply the same meta-learning trick with discriminative approaches? Then it will be helpful to see the contribution to the generative approach.

4- It should include some non-meta learning approaches for comparison like [c].

5-  This approach can be implemented for online continual learning? If yes, then how?

[a]- Meta-Learning Representations for Continual Learning, NeurIPS 2019.

[b]- La-MAML: Look-ahead Meta-Learning for Continual Learning, NeurIPS 2020.

[c]- Online Class-Incremental Continual Learning with Adversarial Shapley Value, AAAI 2021.






**Time Spent Reviewing:**

6

---

> ### Author Response · Authors · 2021-08-09
> **Response to reviewer DcGp**
>
> Thanks for your kind review and helpful comments. To discuss the comments and clarify our work:
>
> 1. As explained in the Related Works section (lines 81-85), our focus is on the methods that use meta-train data to gather experience in an explicit meta-train phase, in order to solve new unseen continual learning problems in the meta-test phase. These approaches contain the OML [a] which you mentioned in your review and we have discussed it deeply in the paper. Other works (Including [b] which you mentioned in your comment and we have also cited in the paper) do not use explicit meta-train phase and focus on solving a single continual-learning problem by taking advantage of meta-learning in the course of solving tasks in this problem. As these methods do not use additional meta-train data as opposed to OML [a], ANML[e], and ours, their experiment settings are not consistent with our approach. So we didn't see it fair to include them in the experiments and comparisons. To make this more clear for readers we will discuss it more in the Related works section.
>
> 2. Following the prior works ([a],[e]) we have only used the Mini-ImageNet and Omniglot datasets which are more common among meta-learning (mainly few-shot) methods. But as you mentioned it is useful to also include CIFAR-100 and we will design experiments on this dataset to include the results in the paper. Recently, we realized that a new paper [d] has reported the performance of the OML and ANML methods on the CIFAR-100 dataset with meta-train and meta-test splits. We ran a similar experiment on CIFAR-100 (Figure 5 of the paper [d]) with our proposed method. Our method achieves an accuracy of $55.6\pm1.2\\%$ for 30-way classification which is approximately $20\\%$ higher than the accuracies reported in Figure 5 of this paper.
>
> 3. As explained in the paper (e.g. lines 234-243) we have experimented discriminative models (MTLR, OML) which use meta-learning to learn representations as we do in the proposed method. We have also proposed a baseline (PGLR) which combines some ideas from generative models with the discriminative approach. Finally comparing IID performance against Continaul (Table-1 main paper) shows how robust these methods are to forgetting. Please let us know if we misunderstood you on the “meta-learning trick”.
>
> 4. Like we mentioned in the first answer and also in the main paper, our method uses explicit meta-training data which is not the case for non-meta-learning methods. With this in mind, we think that it is not fair to compare our results to non-meta-learning methods. For example the paper you suggested [c] has much less accuracy compared to our method on Mini-Imagenet dataset.
>
> 5. In [c], the online continual setting is defined as “we consider the online supervised class-incremental learning setting where a model needs to learn new classes continually from an online data stream (each sample is seen only once).”,  and we described a setting which we called “fully incremental” in the paper as: “However, our proposed method is completely agnostic to the order in which data samples arrive and produces the same model, even in an extreme setting that we refer to as fully incremental. In this setting, only one sample is available at each time step and each data point is visited exactly once” (lines 115-117). With these in mind, we believe that our method is applicable to online continual learning. The detail of applying our method to this setting is also presented in Section 2 of the supplementary materials (Incremental Form for Gaussian Distribution).
>
> Finally thanks for your helpful suggestion on including the block diagram in the main paper, we will surely do that.
>
> [a]- Meta-Learning Representations for Continual Learning, NeurIPS 2019.
>
> [b]- La-MAML: Look-ahead Meta-Learning for Continual Learning, NeurIPS 2020.
>
> [c]- Online Class-Incremental Continual Learning with Adversarial Shapley Value, AAAI 2021.
>
> [d] - Few-Shot and Continual Learning with Attentive Independent Mechanisms, https://arxiv.org/abs/2107.14053
>
> [e] - Learning to Continually Learn, https://arxiv.org/abs/2002.09571

---

> ### Comment · Reviewer_DcGp · 2021-08-20
> **Comments after rebuttal**
>
> I have gone through the rebuttal and found that the authors have answered most of the questions properly. After reading the other's reviews and author rebuttal, I am leaning towards acceptance and keep my rating above the acceptance threshold. I strongly recommend that the authors include the following things in the final revision:
>
> 1-  The authors should include the experimental results for the CIFR-100 dataset.
> 2-  Provide a detailed and proper justification that this model is generic and applies in the online setting without any modification.
> 3- Include all the suggestions provided by other reviewers.

---

### Official Review · Reviewer_vJhk · 2021-07-17

**Rating:** 6
**Confidence:** 4

**Summary:**

This paper proposes a probabilistic generative classifier that is immune to catastrophic forgetting. The proposed lightweight generative classifier achieves high accuracy for new unseen continual learning problems.

**Limitations And Societal Impact:**

The authors adequately addressed the limitations and potential negative societal impact of their work.

**Main Review:**

[Strengths]

The writing of the article is very good.
The method have achieved relatively high classification accuracy. The method of experimental comparison is comprehensive.

[Weaknesses]

In line 140-141, why the generative model is completely immune to catastrophic forgetting, needs a deeper explanation.]

**Time Spent Reviewing:**

5

---

> ### Author Response · Authors · 2021-08-09
> **Response to reviewer vJhk**
>
> Thanks for your helpful review and comments.
> We believe that the mentioned lack of deep explanation about immunity to catastrophic forgetting is because we provided graphical insights (Figure 3) and quantitative results (Table 1) to support this idea in the main paper and left deeper reasoning and justification of the statements for the supplementary materials (Section 1). To resolve this issue and help readers better understand the reason behind the claims, we will add more explanations from the supplementary materials to the main paper in a revised version.
>
> Please let us know if further clarifications are needed.

---

> > ### Comment · Reviewer_vJhk · 2021-09-12
> > **leaning towards acceptance**
> >
> > When reviewing, I have read the first section of the supplementary material and think that this part needs to be more understandable and in-depth.
> > After reading the other's reviews and author rebuttal, I am leaning towards acceptance and keep my rating.

---

### Official Review · Reviewer_UVEJ · 2021-07-20

**Rating:** 5
**Confidence:** 3

**Summary:**

Motivated by biological principles, the authors propose to combine a generative classifier and a meta-learned discriminative embedding network to address meta-continual learning. The results show that the method is more robust against forgetting.

**Limitations And Societal Impact:**

The authors mention on passing positive impact, but not limitations nor potential negative impact.
Note that the claim that lower energy consumption makes the model more "planet-friendly" is usually not true. Actually, it is often the opposite (e.g. rebound effect).

**Main Review:**

Originality:
The idea is moderately novel, since essentialy replaces the prototype classifier used in some methods (eg. iCArL) by a Bayesian version, i.e. a generative classifier. The method is evaluated in the standard benchmarks.

Quality:
The submission seems sound.The claims are supported by several experiments.

Clarity:
The paper is generally well written and clear. The figures as well. However, the authors overemphasize the connections with human learning.

Significance:
The results are positive and state-of-the-art, and the discriminative-generative framework is useful. Another advantage is its computational efficiency and scalability.

Comments:
- Results from other families of meta-learning should be included.
- It is not bd clear the actual connection to biological the authors mention in the motivation, and the mechanism of concept learning.


**Time Spent Reviewing:**

5

---

> ### Author Response · Authors · 2021-08-09
> **Responses to reviewer UVEJ**
>
> Thanks for your constructive feedback.
>
> --**Results from other families of meta-learning**:
>
> To the best of our knowledge, as noted in the Related Works section, the main families of meta-learning are: 1.Optimization-based 2.Metric-based 3.Black box. As our focus is on meta-continual learning, as far as our knowledge goes, the main existing methods are OML [a] and ANML [b], both of which are from the optimization-based family. knowing that our proposed method is related to the metric-based family, the only remaining family is the black box. Since no method from this family has been yet proposed for meta-continual learning we did not include this family in experiments.
>
> --**Actual connection to biological mechanism**:
>
> Most neuroscience studies have analyzed __prototypical learning__ or __Bayesian learning__ at the cognitive or behavioral level. There are also reports providing neural-level evidence about these concepts[c][d], however, the exact neural mechanisms of them are not fully discovered yet. Hence we did not adopt the neural mechanism of these concepts at the circuit level and the connection of our method with these concepts is principally at the cognitive level.
>
> __Prototypical learning__ is a well-studied hypothesis about category learning of the brain [c][e]. Since this type of learning relates to the generative classifiers, we hypothesize that the generative classifiers may play a key role in biological concept learning by providing resistance to catastrophic forgetting. Surprisingly, this approach is overlooked in the literature of continual learning.
>
> __Bayesian inference__ is another aspect of brain functions that is extensively researched [d][f]. Especially in the context of meta-learning, the Bayesian approach offers a natural way to integrate the past experiences in prior knowledge and then update it based on new encountered samples [g].
>
> As we mentioned in line 25, our method fundamentally is built upon the mentioned ideas (prototypical learning and Bayesian inference) and this is the main connection of the proposed method with biological concept learning. To help readers, we will further clarify these connections in the 4.1 section of the revised paper.
>
> --**Energy consumption of the method and its environmental effects**:
>
> Thanks for suggesting the potential drawbacks, our claim for being planet-friendly is backed by Table-6 (appendix), showing that the generative baselines need less time both for train and test phases (and also achieve a higher accuracy) compared to the optimization-based methods on the same hardware. Moreover, more work on parallelization (e.g. larger batch size) can enable the model to take further advantage of more powerful hardwares and reduce the train/test times even more.
>
> It is worth mentioning that a single low-power graphic card is sufficient to reproduce all of the experiments mentioned in the main paper and appendix within a week.
>
>
>
> [a] Meta-learning representations for continual learning. NeurIPS, 2019.
>
> [b] Learning to continually learn, https://arxiv.org/abs/2002.09571
>
> [c] Tracking prototype and exemplar representations in the brain across learning. Elife, 2020.
>
> [d] Bayesian inference with probabilistic population codes. Nature neuroscience, 2006.
>
> [e] Brain mechanisms of concept learning. Journal of Neuroscience, 2019.
>
> [f] The bayesian brain: the role of uncertainty in neural coding and computation. Trends in Neurosciences, 2004.
>
> [g] Spontaneous cortical activity reveals hallmarks of an optimal internal model of the environment. Science, 2011.

---

> ### Comment · Reviewer_UVEJ · 2021-08-28
> **Comments after response from authors**
>
> I appreciate the clarifications. While I believe the paper has useful insights, I also think that it should be more clear on its limitations and connection to other prior works. I still think the connection with biological learning as motivation is somewhat overstated, and the novelty moderate over previous works.
>
> I believe that exemplar-based classifiers (e.g. those in iCaRL) can be seen essentially seen as prototypical classifiers. Every new class is modeled with separate parameters and, for example, the nearest mean classifiers can be seen essentially as generative classifiers. If the backbone is fixed as in the submission, their overall mechanism is very similar. So the paper should be more clear on this close relation with them.
>
> Please also list as limitations the fact that the backbone is not updated. While this prevents forgetting in the backbone by construction, also limits the capacity of learning and beneficial backward transfer from new tasks. In that sense, the paper studies a relatively narrow continual learning setting, since there is no learning in the shared parameters (i.e. backbone). Since there is no catastrophic interference by construction, I'm not sure whether this setting should be considered continual learning (in the line of reviewer TJWB's comment).
>
> Finally, it seems that I wasn't clear enough in my last concern. I don't have any doubt that the method is efficient, so no need to convince me (although that is largely due to only updating the classifier heads). My concern is more general and goes beyond this particular submission, and is with the widely extended yet wrong assumption that being more efficient makes the method more "sustainable/planet-friendly", and therefore being used as motivation/conclusion point. Looking at the big picture, there is an extense literature on the Jevon's paradox and rebound effects, that shows precisely the opposite, i.e. improvements in efficiency, result in more problems for the planet because of increased demand and eventually increased use of natural resources. "More computationally efficient" would be more precise, instead of unnecessarily extrapolating to "more planet-friendly".

---

> > ### Author Response · Authors · 2021-08-31
> > **Responses to reviewer UVEJ**
> >
> > Thanks for your insightful comments.
> >
> > --**connection with biological learning as motivation is somewhat overstated**:
> >
> > The main purpose of presenting biological motivations is that most of the existing continual learning (CL) methods tried to make classifiers more robust to forgetting, just as biological systems. However, they overlooked the point that generative classifiers play an important role in the brain and constructed their methods based on discriminative approaches. Theoretical properties and practical results are good enough to confirm the robustness of the generative classifiers to forgetting, but talking about biological backgrounds encourages the CL community to pay more attention to generative classifiers in order to reach a biological property (robustness to forgetting).
> >
> > Nevertheless, as you suggested, we can describe this motivation in fewer words and we will make it briefer.
> >
> > It is worth mentioning that using a meta-learning approach for CL problems is more compatible with biological lifelong learning. This is also an important issue which we have pointed out very concisely in the paper. A considerable part of neural networks in the brain has a slow learning regime [1,2]. This means that their weights are slowly shaped through life with many different learning problems
> > (meta-learning). The wiring of networks (e.g. ventral stream) in a mature brain negligibly changes when facing a new type of flower or animal[3]! However, the brain also utilizes a fast learning mechanism which increases the learning ability[1,2]. Our method is also biologically plausible in the sense that it uses meta-learning to provide a proper neural network (slow learning), and also fast learning during meta-test which enables learning new concepts with just one repetition.
> >
> > [1] Reinforcement Learning, Fast and Slow. https://www.cell.com/trends/cognitive-sciences/fulltext/S1364-6613(19)30061-0
> >
> > [2] Continual lifelong learning with neural networks: A review. https://www.sciencedirect.com/science/article/pii/S0893608019300231
> >
> > [3] Long-term dendritic spine stability in the adult cortex. https://www.nature.com/articles/nature01276
> >
> > --**I believe that exemplar-based classifiers (e.g. those in iCaRL) can be seen essentially seen as prototypical classifiers….**
> >
> > In short, iCaRL leverages an instance-based memory along with the knowledge distillation technique to combat forgetting. There are some connections between our method and iCaRL especially in the sense that they also worked with prototypes. However, here we discuss some of the differences between these methods which should be taken into account:
> >
> >
> > 1 - As we described in the paper (lines 81 - 83), our method is principally related to "meta-continual learning" which is a new research branch in the field of continual learning [OML]. Hence, Instead of focusing on the common classical setting, the proposed method is well-suited for this specific branch and also considerably improved SOTA.
> >
> > 2 - Our method is not just simply freezing the backbone! The meta-training is an important part of our method to efficiently provide a proper feature extractor for a continual learning problem. This is in line with the motivation of previous works (OML and ANML). Please also note the difference between the accuracy of pretrained and meta-trained backbones (columns 5th and 8th of Table 1). However, iCaRL does not have separate meta-train and meta-test phases to provide a good feature extractor. Freezing the backbone is also not compatible with their method.
> >
> > 3- iCaRL exerts multi-label classification during representation learning and the prototype concept is used only in the final classification phase. Hence, there is an inconsistency between the updating scheme and classification method which may affect the performance.
> >
> > 4 -  iCaRL doesn’t consider inner-class variances in the classification phase while our method is a Bayesian approach and considers a probability distribution over each class based on all of the samples in that class.
> >
> > 5 - Compared with our method, exemplar-based classifiers are not memory efficient and use several times larger memory. We just store two vectors in the low-dimensional embedding space for each category. In contrast, iCaRL stores a considerable number of high-dimensional input examples (at least 20 in experiments) per class in addition to classifier weight vectors. Yet, the prototypes are only formed based on a small proportion of samples of each class which is not representative enough of the whole support set.
> >
> > In general, since exemplar-based classifiers (like K-NN) do not provide a distribution over data samples of each category, they can not be trivially supposed as generative classifiers. Furthermore, exemplar-based methods are potentially more prone to overfitting, especially when a small number of data samples are available.
> >
> > Finally, we believe that the mentioned items make our work distinct enough from iCaRL to be considered valuable. We have also provided insightful comparisons between discriminative and generative classifiers which is certainly beneficial for the both CL and meta-CL community.
> >
> > --**it should be more clear on its limitations and connection to other prior works**
> >
> > We did not discuss the mentioned detailed differences in the paper to prevent  "Related Works" from getting too long and out of standard. But it seems necessary to add more clarifications about connections with exemplar-based methods for readers.
> >
> > --**list as limitations the fact that the backbone is not updated**
> >
> > We have pointed out this limitation in the Discussion section (lines 294-298).
> >
> > --**the paper studies a relatively narrow continual learning setting, since there is no learning in the shared parameters**
> >
> > Note that our experiment settings are the same as previous meta-continual learning methods (OML, ANML), moreover our method does not need to know about the total number of classes per episode which is needed in prior works. For discussion on learning in the shared parameters and forward/backward transfer please refer to our response to the reviewer TJWB.
> >
> > “To the best of our knowledge, the main purpose of continual learning is to alleviate catastrophic forgetting and the positive forward/backward transfers are additional desired properties. … “
> >
> > --**although that is largely due to only updating the classifier heads**
> >
> > --**More computationally efficient" would be more precise, instead of unnecessarily extrapolating to "more planet-friendly**
> >
> > Thank you for your deep description. As you suggested, to become more precise, we will replace the "more planet-friendly" with "more computationally efficient".
> >
> > Please also note that the main reason for the efficiency of the proposed method is not just updating the classifier heads. This is true when comparing meta-testing with meta-training. however, Table 6 (appendix) shows that our method is also computationally efficient in meta-training. The main reason for this efficiency is the type of optimization in our method, which is not second-order (as opposed to OML and ANML).

---

### Decision · Program_Chairs · 2021-09-27

**Decision:**

Accept (Poster)

**Comment:**

This paper addresses catastrophic forgetting in continual learning using inspirations from cognitive and neuroscience. The drawbacks of discriminative models are first demonstrated, and then a probabilistic generative method is presented, leveraging prototypical learning by Bayesian approach.  Experimental results confirm the effectiveness of the new method.

All reviewers and I find the paper interesting and well written.  The new method is efficient and effective in practice, and the intuition is well illustrated.  The rebuttal well addressed some concerns, and the authors are encouraged to incorporate them into the final version.